# Learning to Solve Differential Equation Constrained Optimization Problems

**Vincenzo Di Vito**[1]**, Mostafa Mohammadian** [2]**, Kyri Baker** [2]**, Ferdinando Fioretto** [1]

[1] Department of Computer Science, University Of Virginia,
{eda8pc,fioretto}@virginia.edu
[2] College of Engineering and Applied Science, University of Colorado Boulder,
{mostafa.mohammadian,kyri}@colorado.edu

## Abstract

Differential equations (DE) constrained optimization plays a critical role in numerous scientific and engineering fields, including energy systems, aerospace engineering, ecology, and finance, where optimal configurations or control strategies must be determined for systems governed by ordinary or stochastic differential equations. Despite its significance, the computational challenges associated with these problems have limited their practical use. To address these limitations, this paper introduces a learning-based approach to DE-constrained optimization that combines techniques from *proxy optimization* (Kotary et al., 2021) and *neural differential equations* (Chen et al., 2018). The proposed approach uses a dual-network architecture, with one approximating the control strategies, focusing on steady-state constraints, and another solving the associated DEs. This combination enables the approximation of optimal strategies while accounting for dynamic constraints in near real-time. Experiments across problems in energy optimization and finance modeling show that this method provides full compliance with dynamic constraints and it produces results up to 25 times more precise than other methods which do not explicitly model the system's dynamic equations.

## 1 Introduction

In a wide array of scientific and engineering applications, differential equations (DEs) serve as a fundamental tool to model dynamic phenomena where precise predictions and optimal control are crucial. These applications range from energy optimization, where generator dynamics are required to assess system stability, to aerospace engineering, relying on trajectory optimization, and finance, where asset price prediction hinges on stochastic processes. Central to these applications is the optimization of systems constrained by Ordinary (ODEs) or Stochastic (SDEs) Differential Equations, referred to as *DE-constrained optimization problems*. These problems entail not only solving the DEs but also optimizing decision variables subject to the dynamics dictated by these equations.

This dual requirement, however, poses significant computational challenges. Traditional approaches, such as shooting methods (Gerdts, 2003), collocation methods (Fairweather & Meade, 2020), and discretization techniques (Betts & Campbell, 2005), are known to struggle with scalability and precision, especially on high-dimensional and nonlinear systems, which are of interest in this paper. To address these challenges, this paper introduces a novel learning-based DE-optimization proxy that integrates advancements from two key methodologies: *proxy optimizers* (Kotary et al., 2021) and *neural differential equations* (neural-DEs) (Chen et al., 2018; Kidger, 2022). In our approach, a neural network serves as a proxy optimizer, approximating solutions to the decision problem while simultaneously leveraging another neural network to solve the underlying DEs. This novel dual-network architecture exploits a primal-dual method to ensure that both the dynamics dictated by the DEs and the optimization objectives are concurrently learned and respected. Importantly, this integration allows for end-to-end differentiation enabling efficient gradient-based optimization.

The proposed method is validated across two domains: energy systems and financial modeling and optimization. The experimental results show the ability to directly handle DE in the optimization surrogate, which allows our method to produce solutions that are up to 25 times more precise than

standard proxy optimization techniques, while also adhering to system dynamics. *This precision improvement is important and opens new avenues for research and application in fields that demand high-fidelity dynamic modeling and optimal decision-making at low computational budgets.*

**Contributions.** The paper makes the following contributions: **(1)** It introduces a novel learning-based method to efficiently approximate solutions of DE-constrained optimization problems. Our approach is a unique integration of neural-DE models to capture the system dynamics and proxy optimizers to approximate the problem decision variables. These components are sinergistically integrated into model training via a primal-dual method. **(2)** It empirically demonstrates the importance of incorporating the system dynamics into an optimization framework, by showing that proxy optimizers methods who neglect these dynamic behaviors systematically violate the DE requirements. **(3)** It shows that capturing the system dynamics with neural differential surrogate models, leads up to 25 times higher solution quality compared to other learning-based approaches capturing the dynamics such as PINN (Raissi et al., 2019) or LSTM (Yu et al., 2019).

## 2 RELATED WORKS

Model Predictive Control (MPC) (Mayne et al., 2000) is a key control system technique that predicts future states over a time horizon to determine optimal control actions while satisfying constraints. This involves solving an optimization problem using predicted states. Alternatively, DAE-constrained optimization (Blajer & Kołodziejczyk, 2004) directly incorporates the system's differential and algebraic equations into the optimization problem via discretization, transforming it into a finite-dimensional form. However, while retaining dependence on independent variables, this increases problem dimensionality. The nonlinearities in both optimization and system dynamics, along with the need for real-time solutions, make these approaches impractical in our setting.

In recent years, there has been growing interest in leveraging neural network architectures to approximate solutions of challenging constrained optimization problems. Termed *proxy optimizers*, these methods create fast surrogate models by learning mappings from optimization problem parameters to optimal solutions (Kotary et al., 2021). Some approaches rely on supervised learning with precomputed solutions (Fioretto et al., 2020), while others employ self-supervised strategies by exploiting the problem's structure (Park & Van Hentenryck, 2023). A major challenge in this setting is constraint satisfaction, often tackled through penalty-based methods (Fioretto et al., 2020), implicit layers (Donti et al., 2020), or efficient post-inference projection techniques (Kotary et al., 2024). Additionally, neural networks have been applied to approximate solutions of differential equations. Physics-Informed Neural Networks (PINNs) (Raissi et al., 2019) integrate physical laws within their architecture but face challenges in training and generalization (Wang et al., 2020; Kovachki et al., 2024). Neural differential equations (Kidger, 2022; Chen et al., 2018) provide an alternative by modeling system dynamics through parameterized hidden state derivatives, which allows to capture complex dynamics effectively from data observations.

Our work builds on these areas for surrogate modeling and introduces a learning-based approach to solve, for the first time to our knowledge, DE-constrained optimization problems in near real-time.

## 3 SETTINGS AND GOALS

Consider an optimization problem constrained by a system of ordinary differential equations [1]:

$$\underset{\boldsymbol{u}}{\text{Minimize}} \quad \overbrace{L(\boldsymbol{u}, \boldsymbol{y}(T)) + \int_{t=0}^{T} \Phi(\boldsymbol{u}, \boldsymbol{y}(t), t)\, dt}^{\mathcal{J}(\boldsymbol{u}, \boldsymbol{y}(t))} \tag{1a}$$

$$\texttt{s.t.} \quad d\boldsymbol{y}(t) = \boldsymbol{F}(\boldsymbol{u}, \boldsymbol{y}(t), t)dt \tag{1b}$$

$$\boldsymbol{y}(0) = \boldsymbol{I}(\boldsymbol{u}) \tag{1c}$$

$$\boldsymbol{g}(\boldsymbol{u}, \boldsymbol{y}(t)) \leq 0; \quad \boldsymbol{h}(\boldsymbol{u}, \boldsymbol{y}(t)) = 0, \tag{1d}$$

where $\boldsymbol{u} = (u_1, \ldots, u_n)$ represents the vector of decision variables and $\boldsymbol{y}(t) = (y_1(t), \ldots, y_m(t))$ denotes the state variables, each governed by a differential equation $dy_i(t) = F_i(\boldsymbol{y}(t), \boldsymbol{u}, t)dt$. Here

---

[1]to ease notation the paper focuses on ODEs, and refers the reader to Appendix A for an extension to SDE

each $F_i$ describes the dynamic behavior of the system through ODEs. The set of all such ODEs is captured by $\boldsymbol{F}$, as defined in Constraint (1b). Note that these DEs are parametrized by decision variables $\boldsymbol{u}$, rendering the coupling between the control strategy and the system's dynamic response highly interdependent. The objective function $\mathcal{J}$ (1a) aims to minimize a combination of the running cost $\Phi$, which varies with the state and decision variables over time, and the terminal cost $L$, which depends on the final state $\boldsymbol{y}(T)$ and the decision variables $\boldsymbol{u}$. The time horizon $T$ defines the period over which the optimization takes place. Constraint (1c) sets the initial conditions for the state variables based on the decision variables $\boldsymbol{u}$. Additional constraints (1d) enforce sets of inequality and equality conditions on the state and decision variables, ensuring that the system constraints are met throughout the decision process.

**Energy system example.** For example, in the context of power system optimization, decision variables $\boldsymbol{u}$ capture generators' power outputs, and state variables $\boldsymbol{y}(t)$ describe generator rotor angles and speeds, which are key for system stability. The system dynamics in $\boldsymbol{F}$, capture the electro-mechanical interactions in the power network, and their initial conditions, as determined by (1c), are set based on the decision variables. The objective function $\mathcal{J}$ aims to minimize immediate operational costs like fuel consumption ($\Phi$) and address long-term

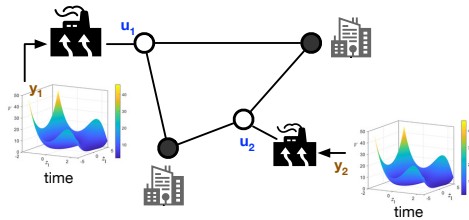

Figure 1: Decision variables $u$ represent generators outputs, which are influenced by state variables $y$ describing rotor angles and speed.

costs ($L$) over a specific time horizon $T$. Optimizing the generator outputs is finally subject to engineering operational limits and physics constraints (Constraints (1d) , e.g. Ohm's and Kirkhoff's laws). An illustration is provided in Figure 1 and the problem description in Appendix B.

## CHALLENGES

While being fundamental for many applications, Problem (1) presents three key challenges:
1. Finding optimal solutions to Problem (1) is computationally intractable. Even without the differential equation constraints, the decision version of the problem alone is NP-hard in general.
2. Achieving high-quality approximations of the system dynamics (Equations 1b) and (1c) in near real-time, poses the second significant challenge. The high dimensionality and non-linearity of these dynamics further complicate the task.
3. Finally, the integration of the system dynamics into the decision-making process for solving Problem (1) poses another challenge. Indeed, including differential equations (1b) in the optimization framework renders traditional numerical methods impractical for real-time applications.

The next section focuses on providing a solution to each of these challenges.

## 4    DE-OPTIMIZATION PROXY

To address the challenges outlined above, the paper introduces *DE-Optimization Proxy* (DE-OP): a fully differentiable DE-optimization surrogate. In a nutshell, DE-OP defines a dual-network architecture, where one neural network, named $\mathcal{F}_\omega$, approximates the *optimal* decision variables $\boldsymbol{u}^*$, and another, denoted as $\mathcal{N}_\theta$, approximates the associated state variables $\boldsymbol{y}(t)$, based on the concept of neural differential equations. Here $\omega$ and $\theta$ represents the models' $\mathcal{F}_\omega$ and $\mathcal{N}_\theta$ trainable parameters, respectively. An illustration of the overall framework and the resulting interaction of the dual network is provided in Figure 2. The subsequent discussion first describes these two components individually and then shows their integration by exploiting a primal-dual learning framework.

**Optimizing over distribution of instances.** To enable a learnable mechanism for addressing DE-constrained optimization, DE-OP operates over a distribution $\Pi$ of problem instances induced by problem parameters $\boldsymbol{\zeta}$, and aims to train a model across this distribution. The learning framework takes problem parameters $\boldsymbol{\zeta}$ as inputs and generates outputs $\hat{\boldsymbol{u}}$, representing an approximation of the optimal decision variables while adhering to the constraint functions (1b)–(1d). With reference to

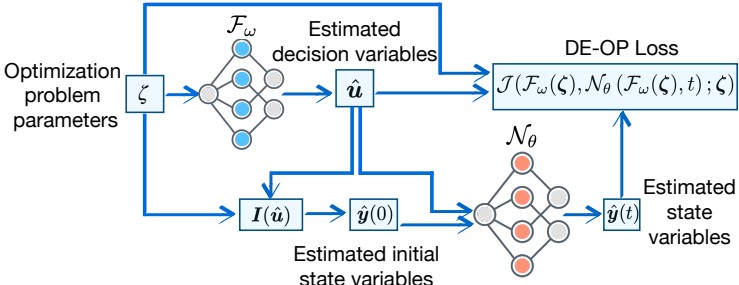

Figure 2: DE-OP uses a dual network architecture consisting of a proxy optimization model $\mathcal{F}_\omega$ to estimate the decision variables $\hat{\boldsymbol{u}}$ and a neural-DE model $\mathcal{N}_\theta$ to estimate the state-variables $\hat{\boldsymbol{y}}(t)$, with the objective function $\mathcal{J}(\mathcal{F}_\omega(\boldsymbol{\zeta}), \mathcal{N}_\theta(\mathcal{F}_\omega(\boldsymbol{\zeta}), t); \boldsymbol{\zeta})$ capturing the overall loss.

(1a), the formal learning objective is:

$$\underset{\omega, \theta}{\text{Minimize}}\, \mathbb{E}_{\boldsymbol{\zeta} \sim \Pi}\left[\mathcal{J}\left(\mathcal{F}_\omega(\boldsymbol{\zeta}), \mathcal{N}_\theta(\mathcal{F}_\omega(\boldsymbol{\zeta}), t); \boldsymbol{\zeta}\right)\right] \tag{2a}$$

$$\texttt{s.t.}\quad \text{(1b)–(1d)}, \tag{2b}$$

where $\hat{\boldsymbol{u}} = \mathcal{F}_\omega(\boldsymbol{\zeta})$ and the state variables estimate is denoted as $\hat{\boldsymbol{y}}(t) = \mathcal{N}_\theta(\mathcal{F}_\omega(\boldsymbol{\zeta}), t)$. Here, $\boldsymbol{\zeta}$ parameterizes each problem instance, representing constants such as customer demands in the power system example. Although the problem structure remains consistent across instances, each one involves a distinct decision problem, leading to unique state variable trajectories. Given the complexity of solving Problem (2), the goal is to develop a fast and accurate neural DE optimization surrogate. This approach uses the concept of *proxy optimizers*, which is detailed next.

## 4.1 NEURAL OPTIMIZATION SURROGATE

The first objective is to establish a neural network-based mapping $F : \Pi \to \mathcal{U}$ that transforms parameters $\boldsymbol{\zeta} \sim \Pi$ from a DE-constrained optimization problem (1) into optimal decisions $\boldsymbol{u}^\star(\boldsymbol{\zeta}) \in \mathcal{U}$, operating under the restriction that $T = 0$, (i.e. the dynamics of the system are absent). Practically, this mapping is modeled as a neural network $\mathcal{F}_\omega$ which learns to predict the optimal decision variables from the problem parameters. The model training uses a dataset $\mathcal{D} = \{(\boldsymbol{\zeta}_i, \boldsymbol{u}_i^\star)\}_{i=1}^N$, of $N$ samples, with each sample $(\boldsymbol{\zeta}_i, \boldsymbol{u}_i^\star)$ including the observed problem parameters $\boldsymbol{\zeta}_i$ and the corresponding (steady-state) optimal decision variables $\boldsymbol{u}_i^\star$. The training objective is to refine $\mathcal{F}_\omega$ in a way that it closely approximates the ideal mapping $F$. Several approaches have been proposed to build such surrogate optimization solvers, many of which leverage mathematical optimization principles (Fioretto et al., 2020; Park & Van Hentenryck, 2023) and implicit layers (Donti et al., 2020), to encourage or ensure constraint satisfaction (see Appendix C.1 for an in-depth discussion of such methods). Once trained, the model $\mathcal{F}_\omega$ can be used to generate near-optimal solutions at low inference times. We leverage this idea to learn the mapping $F$. As shown in Figure 2, the estimated optimal decisions $\hat{\boldsymbol{u}} = \mathcal{F}_\omega(\boldsymbol{\zeta})$ are then fed to a neural-DE model, which will be discussed next.

## 4.2 NEURAL ESTIMATION OF THE STATE VARIABLES

The second objective of DE-OP involves to efficiently capture the system dynamics of DE-constrained optimization problems. This is achieved by developing neural DE models $\mathcal{N}_\theta$ to learn solutions of a *parametric family* of differential equations (Kidger, 2022). *Since these DE-surrogates are fully differentiable, they are particularly suitable for integration with the optimization surrogate introduced in the previous section*, aligning with our goal defined in Equation (2).

The optimization proxy estimated solutions $\hat{\boldsymbol{u}}$ determine the initial state variables $\boldsymbol{y}(0)$ through the function $\boldsymbol{I}$: $\hat{\boldsymbol{y}}(0) = \boldsymbol{I}(\hat{\boldsymbol{u}})$. As shown in Figure 2, given a solution $\hat{\boldsymbol{u}}$, a neural-DE model $\mathcal{N}_\theta$ generates an estimate $\hat{\boldsymbol{y}}$ of the state variables that satisfies:

$$d\hat{\boldsymbol{y}}(t) = \mathcal{N}_\theta(\hat{\boldsymbol{u}}, t)dt \tag{3a}$$

$$\hat{\boldsymbol{y}}(0) = \boldsymbol{I}(\hat{\boldsymbol{u}}). \tag{3b}$$

The remainder of this section details the methods by which the state variables are precisely estimated using a Neural Differential Equation model.

**Model initialization and training.** Given that the neural-DE model $\mathcal{N}_\theta$ takes as input the estimated decision variables $\hat{\boldsymbol{u}}$ provided by the DE-OP's optimization proxy $\mathcal{F}_\omega$, where $\hat{\boldsymbol{u}} = \mathcal{F}_\omega(\boldsymbol{\zeta})$, it is practical to initialize (or hot-start) the neural-DE model effectively. To achieve this, we construct a dataset $\mathcal{D} = \{(\boldsymbol{u}'_i, \boldsymbol{y}^\star_i(t))\}_{i=1}^N$, where each $\boldsymbol{u}'_i$ is a near-optimal decision sampled within the bounds specified by constraints $\boldsymbol{g}$ (e.g., $u'_j \sim \mathcal{U}(a_j, b_j)$ if $u_j$'s bound is defined by $a_j$ and $b_j$). The corresponding state trajectories $\boldsymbol{y}^\star_i(t)$ are obtained by numerically solving the differential equations with initial condition $\boldsymbol{y}(0) = \boldsymbol{I}(\boldsymbol{u}'_i)$. The neural-DE model is trained by minimizing the loss:

$$\underset{\theta}{\text{Minimize}}\, \mathbb{E}_{(\boldsymbol{x}, \boldsymbol{y}) \sim \mathcal{D}} \left[ \|\mathcal{N}_\theta(\boldsymbol{x}, t) - \boldsymbol{y}(t)\|^2 \right], \tag{4}$$

where $\boldsymbol{x} = \boldsymbol{u}'$ and $\hat{\boldsymbol{y}}(t) = \mathcal{N}_\theta(\boldsymbol{x}, t)$.

Since $\mathcal{F}_\omega$ approximates $\boldsymbol{u}^\star$, it may introduce errors. To mitigate this, the dataset $\mathcal{D}$ is constructed using near-optimal decisions $\boldsymbol{u}'$ sampled from the feasible bounds $\boldsymbol{g}$, ensuring that $\mathcal{N}_\theta$ is trained on a distribution $\Pi_{\boldsymbol{u}'} \approx \Pi_{\hat{\boldsymbol{u}}}$. This approach does not require exact optimal decisions and assumes only small estimation errors from $\mathcal{F}_\omega$, which is typically valid in practice.

Once trained, the neural-DE model $\mathcal{N}_\theta$ can accurately estimate state variables $\hat{\boldsymbol{y}}(t)$ for decisions $\hat{\boldsymbol{u}}$ produced by $\mathcal{F}_\omega$. This integration enables end-to-end training of the DE-OP framework, ensuring that both decision and state variables are optimized cohesively. While other learning-based methods, such as Physics-Informed Neural Networks (PINN) (Raissi et al., 2019) and Long Short-Term Memory (LSTM) models (Yu et al., 2019), can be used to efficiently estimate the state variables, due to a generalization bias (PINN), and lack of dynamic equations modeling (LSTM), they produce substantially less precise predictions than the neural-DE models. For a comparison of neural-DE models with these alternative approaches across the experimental tasks described in Section 5, please refer to Appendices D.1 and E.1.

### 4.3 HANDLING STATIC AND DYNAMICS CONSTRAINTS JOINTLY

To integrate the neural-DE models within the decision process, this paper proposes a Lagrangian Dual (LD) learning approach, which is inspired by the generalized augmented Lagrangian relaxation technique (Hestenes, 1969) adopted in classic optimization. In Lagrangian relaxation, some or all the problem constraints are relaxed into the objective function using Lagrangian multipliers to capture the penalty induced by violating them. The proposed formulation leverages Lagrangian duality to integrate trainable and weighted regularization terms that encapsulate both the static and dynamic constraints violations. When all the constraints are relaxed, the violation-based Lagrangian relaxation of problem (1) defines as

$$\underset{\boldsymbol{u}}{\text{Minimize}}\, \mathcal{J}(\boldsymbol{u}, \boldsymbol{y}(t)) + \boldsymbol{\lambda}_{h'}^\top |\boldsymbol{h}'(\boldsymbol{u}, \boldsymbol{y}(t))| + \boldsymbol{\lambda}_g^\top \max(0, \boldsymbol{g}(\boldsymbol{u}, \boldsymbol{y}(t))),$$

where $\mathcal{J}, \boldsymbol{g}$ are defined in (1a), and (1d), respectively, and $\boldsymbol{h}'$ is defined as follows,

$$\boldsymbol{h}'(\boldsymbol{u}, \boldsymbol{y}(t)) = \begin{bmatrix} d\boldsymbol{y}(t) - \boldsymbol{F}(\boldsymbol{u}, \boldsymbol{y}(t), t)dt \\ \boldsymbol{y}(0) - \boldsymbol{I}(\boldsymbol{u}) \\ \boldsymbol{h}(\boldsymbol{u}, \boldsymbol{y}(t)). \end{bmatrix} \tag{5}$$

It denotes the set of *all* equality constraints of problem (1), thus extending the constraints $\boldsymbol{h}$ in (1d), with the system dynamics (1b) and the initial conditions equations (1c) written in an implicit form as above. Therein, $\boldsymbol{\lambda}_{h'}$ and $\boldsymbol{\lambda}_g$ are the vectors of Lagrange multipliers associated with functions $\boldsymbol{h}'$ and $\boldsymbol{g}$, e.g. $\lambda_{h'}^i, \lambda_g^j$ are associated with the $i$-th equality $h'_i$ in $\boldsymbol{h}'$ and $j$-th inequality $g_j$ in $\boldsymbol{g}$, respectively. The key advantage of expressing the system dynamics (1b) and initial conditions (1c) in the same implicit form as the equality constraints $\boldsymbol{h}$, (as shown in (5)), is that the system dynamics can be treated in the same manner as the constraint functions $\boldsymbol{h}$. This enables us to satisfy the system dynamics and the static set of constraints ensuring that they are incorporated seamlessly into the optimization process.

The proposed primal-dual learning method uses an iterative approach to find good values of the *primal* $\omega, \theta$ and dual $\boldsymbol{\lambda}_{h'}, \boldsymbol{\lambda}_g$ variables; it uses an augmented modified Lagrangian as a loss function to train the prediction $\hat{\boldsymbol{u}}, \hat{\boldsymbol{y}}(t)$ as employed

$$\mathcal{L}^{\text{DE-OP}}(\hat{\boldsymbol{u}}, \boldsymbol{u}^\star, \hat{\boldsymbol{y}}(t)) = \|\hat{\boldsymbol{u}} - \boldsymbol{u}^\star\|^2 + \boldsymbol{\lambda}_{h'}^\top |\boldsymbol{h}'(\hat{\boldsymbol{u}}, \hat{\boldsymbol{y}}(t))| + \boldsymbol{\lambda}_g^\top \max(0, \boldsymbol{g}(\hat{\boldsymbol{u}}, \hat{\boldsymbol{y}}(t))), \tag{6}$$

---

**Algorithm 1** Primal Dual Learning for DE-Constrained Optimization

---

1: **Input:** Dataset $\mathcal{D} = \{(\boldsymbol{\zeta}_i, \boldsymbol{u}_i^\star)\}_{i=1}^N$; optimizer method, learning rate $\eta$ and Lagrange step size $\rho$.
2: Initialize Lagrange multipliers $\boldsymbol{\lambda}_{h'}^0 = 0$, $\boldsymbol{\lambda}_g^0 = 0$.
3: **For** each epoch $k = 0, 1, 2, \dots$
4:     **For** each $(\boldsymbol{\zeta}_i, \boldsymbol{u}_i^\star) \in \mathcal{D}$
5:         $\hat{\boldsymbol{u}}_i \leftarrow \mathcal{F}_{\omega^k}(\boldsymbol{\zeta}_i), \quad \hat{\boldsymbol{y}}_i(t) \leftarrow \mathcal{N}_{\theta^k}(\mathcal{F}_{\omega^k}(\boldsymbol{\zeta}_i), t)$
6:         Compute loss function: $\mathcal{L}^{\text{DE-OP}}(\hat{\boldsymbol{u}}_i, \boldsymbol{u}_i^\star, \hat{\boldsymbol{y}}_i(t))$ using (6)
7:         Update DE-OP model parameters:

$$\omega^{k+1} \leftarrow \omega^k - \eta \nabla_\omega \mathcal{L}^{\text{DE-OP}}\left(\mathcal{F}_{\omega^k}^{\boldsymbol{\lambda}^k}(\boldsymbol{\zeta}), \boldsymbol{u}^\star, \mathcal{N}_{\theta^k}^{\boldsymbol{\lambda}^k}\left(\mathcal{F}_{\omega^k}^{\boldsymbol{\lambda}^k}(\boldsymbol{\zeta}), t\right)\right)$$

$$\theta^{k+1} \leftarrow \theta^k - \eta \nabla_\theta \mathcal{L}^{\text{DE-OP}}\left(\mathcal{F}_{\omega^k}^{\boldsymbol{\lambda}^k}(\boldsymbol{\zeta}), \boldsymbol{u}^\star, \mathcal{N}_{\theta^k}^{\boldsymbol{\lambda}^k}\left(\mathcal{F}_{\omega^k}^{\boldsymbol{\lambda}^k}(\boldsymbol{\zeta}), t\right)\right)$$

8:     Update Lagrange multipliers:

$$\boldsymbol{\lambda}_{h'}^{k+1} \leftarrow \boldsymbol{\lambda}_{h'}^k + \rho|\boldsymbol{h}'(\hat{\boldsymbol{u}}, \hat{\boldsymbol{y}}(t))|, \qquad \boldsymbol{\lambda}_g^{k+1} \leftarrow \boldsymbol{\lambda}_g^k + \rho \max(0, \boldsymbol{g}(\hat{\boldsymbol{u}}, \hat{\boldsymbol{y}}(t))).$$

---

where $\|\hat{\boldsymbol{u}} - \boldsymbol{u}^\star\|^2$ represents the decision error, with respect to *steady-state* optimal decision $\boldsymbol{u}^\star$, while $\boldsymbol{\lambda}_{h'}^\top |\boldsymbol{h}'(\hat{\boldsymbol{u}}, \hat{\boldsymbol{y}}(t))|$ and $\boldsymbol{\lambda}_g^\top \max(0, \boldsymbol{g}(\hat{\boldsymbol{u}}, \hat{\boldsymbol{y}}(t)))$ measures the constraint violations incurred by prediction $\hat{\boldsymbol{u}} = \mathcal{F}_\omega(\boldsymbol{\zeta})$ and $\hat{\boldsymbol{y}}(t) = \mathcal{N}_\theta(\mathcal{F}_\omega(\boldsymbol{\zeta}), t)$. This accounts for the contribution of both networks $\mathcal{F}, \mathcal{N}$ during training, which is balanced via iterative updates of the Lagrange multipliers in (6) based on the amount of violation of the associated constraint function. To ease notation, in Equation (6) the dependency from parameters $\boldsymbol{\zeta}$ is omitted. At iteration $k + 1$, finding the optimal parameters $\omega, \theta$ requires solving

$$\omega^{k+1}, \theta^{k+1} = \arg\min_{\omega, \theta} \mathbb{E}_{(\boldsymbol{\zeta}, \boldsymbol{u}^\star) \sim \mathcal{D}}\left[\mathcal{L}^{\text{DE-OP}}\left(\mathcal{F}_\omega^{\boldsymbol{\lambda}^k}(\boldsymbol{\zeta}), \boldsymbol{u}^\star, \mathcal{N}_\theta^{\boldsymbol{\lambda}^k}\left(\mathcal{F}_\omega^{\boldsymbol{\lambda}^k}(\boldsymbol{\zeta}), t\right)\right)\right],$$

where $\mathcal{F}_\omega^{\boldsymbol{\lambda}^k}$ and $\mathcal{N}_\theta^{\boldsymbol{\lambda}^k}$ denote the DE-OP's optimization and predictor models $\mathcal{F}_\omega$ and $\mathcal{N}_\theta$, at iteration $k$, with $\boldsymbol{\lambda}^k = [\boldsymbol{\lambda}_{h'}^k \ \boldsymbol{\lambda}_g^k]^\top$. This step is approximated using a stochastic gradient descent method

$$\omega^{k+1} = \omega^k - \eta \nabla_\omega \mathcal{L}^{\text{DE-OP}}\left(\mathcal{F}_{\omega^k}^{\boldsymbol{\lambda}^k}(\boldsymbol{\zeta}), \boldsymbol{u}^\star, \mathcal{N}_{\theta^k}^{\boldsymbol{\lambda}^k}\left(\mathcal{F}_{\omega^k}^{\boldsymbol{\lambda}^k}(\boldsymbol{\zeta}), t\right)\right)$$

$$\theta^{k+1} = \theta^k - \eta \nabla_\theta \mathcal{L}^{\text{DE-OP}}\left(\mathcal{F}_{\omega^k}^{\boldsymbol{\lambda}^k}(\boldsymbol{\zeta}), \boldsymbol{u}^\star, \mathcal{N}_{\theta^k}^{\boldsymbol{\lambda}^k}\left(\mathcal{F}_{\omega^k}^{\boldsymbol{\lambda}^k}(\boldsymbol{\zeta}), t\right)\right),$$

where $\eta$ denotes the learning rate and $\nabla_\omega \mathcal{L}$ and $\nabla_\theta \mathcal{L}$ represent the gradients of the loss function $\mathcal{L}$ with respect to the parameters $\omega$ and $\theta$, respectively, at the current iteration $k$. Importantly, this step does not recomputes the training parameters from scratch, but updates the weights $\omega, \theta$ based on their value at the previous iteration. Finally, the Lagrange multipliers are updated as

$$\boldsymbol{\lambda}_{h'}^{k+1} = \boldsymbol{\lambda}_{h'}^k + \rho|\boldsymbol{h}'(\hat{\boldsymbol{u}}, \hat{\boldsymbol{y}}(t))|$$
$$\boldsymbol{\lambda}_g^{k+1} = \boldsymbol{\lambda}_g^k + \rho \max(0, \boldsymbol{g}(\hat{\boldsymbol{u}}, \hat{\boldsymbol{y}}(t))),$$

where $\rho$ denotes the Lagrange step size. The overall training scheme is presented in Algorithm 1. It takes as input the training dataset $\mathcal{D} = \{(\boldsymbol{\zeta}_i, \boldsymbol{u}_i^\star)\}_{i=1}^N$, the learning rate $\eta > 0$, and the Lagrange step size $\rho > 0$. The Lagrange multipliers are initialized in line 2. As shown in Figure 2, for each sample in the dataset (line 4), the DE-OP's optimization model $\mathcal{F}_{\omega^k}$ computes the predicted decisions $\hat{\boldsymbol{u}}_i$, while $\mathcal{N}_\theta$ computes an estimate of the state variables $\hat{\boldsymbol{y}}_i(t)$ (line 5). The loss function $\mathcal{L}^{\text{DE-OP}}$ is computed (line 6) incorporating both the objective and the constraints and using the predicted values $\hat{\boldsymbol{u}}, \hat{\boldsymbol{y}}(t)$ and the Lagrange multipliers $\boldsymbol{\lambda}_{h'}^k$ and $\boldsymbol{\lambda}_g^k$. The weights $\omega, \theta$ of the DE-OP models $\mathcal{F}_\omega, \mathcal{N}_\theta$ are then updated using stochastic gradient descent (SGD) (line 7). Finally, at the end of the epoch, the Lagrange multipliers are updated based on the respective constraints violations (line 8).

While the DE-OP model training algorithm is described extending the Lagrangian Dual Learning approach (Fioretto et al., 2020), the flexibility of DE-OP allows to leverage other proxy optimizer methods, such as the self-supervised Primal-Dual Learning (Park & Van Hentenryck, 2023), which could similarly be extended to integrate the system dynamics via neural-DE modeling within the DE-OP framework, as the experiments will show. When near-optimal solutions $\boldsymbol{u}^\star$ are not available, the term $\|\hat{\boldsymbol{u}} - \boldsymbol{u}^\star\|^2$ in Equation 6 can be replaced with $\mathcal{J}(\boldsymbol{u}, \boldsymbol{y}(t))$ to facilitate self-supervised learning, which has shown promising results (Donti et al., 2020) in a variety of tasks.

## 5 EXPERIMENTAL SETTING

This section evaluates the DE-OP model on a financial modeling and energy optimization tasks. Given the absence of other methods capable of meeting the stringent time requirements for solving DE-constrained optimization problems, we compare several proxy optimizer methods as baselines. However, these baselines focus on the "steady-state" aspects of the problem by omitting the system dynamic components, such as the objective term $\Phi$ and the system dynamic constraints (1b) and (1c) from Problem (1). They aim to approximate the optimal decision variables $\hat{\boldsymbol{u}}$ to $\boldsymbol{u}^*$ that could be obtained if the system was at a steady-state. We evaluate the Lagrangian Dual approach (**LD**) of Fioretto et al. (2020), which uses a penalty-based method for constraint satisfaction, Deep Constraint Completion and Correction (**DC3**) from Donti et al. (2020) that enforces constraint satisfaction through a completion-correction technique, self-supervised learning (**PDL**) of Park & Van Hentenryck (2023) using an augmented Lagrangian loss function, and a method (**MSE**) that minimizes the mean squared error between the predicted solutions $\hat{\boldsymbol{u}}$ and the pre-computed (steady-state) solutions $\boldsymbol{u}^*$. A comprehensive description of these methods is provided in Appendix C.1.

Furthermore, the comparison includes various learning-based DE-surrogate solvers in place of the network $\mathcal{N}_\theta$ in our framework, including neural-differential equations Kidger (2022), PINNs Raissi et al. (2019), and LSTM networks. The experiments focus on two main aspects: **(1)** comparing DE-OP with proxy optimizers that capture only the steady-state problems, focusing on the system dynamics violations, and **(2)** assessing the effectiveness of the various surrogate DE-solver methods.

### 5.1 DYNAMIC PORTFOLIO OPTIMIZATION

The classical Markowitz Portfolio Optimization (Rubinstein, 2002), described by (9a)-(9c), consists of determining the investment allocations within a portfolio to maximize a balance of return and risk. The paper extends this task by incorporating the stochastic dynamic (9d) of the asset prices, based on a simplified Black-Scholes model (Capiński & Kopp, 2012). This model represents a real-world scenario where asset prices fluctuates, and investment decisions are made in advance, such as at the market's opening, based on the final asset prices forecast. The task defines as:

$$\underset{\boldsymbol{u}}{\text{Minimize}} \quad \mathbb{E}\left[-\boldsymbol{y}(T)^\top \boldsymbol{u} + \boldsymbol{u}^\top \boldsymbol{\Sigma} \boldsymbol{u}\right] \tag{9a}$$

$$\text{s.t.} \quad \mathbf{1}^\top \boldsymbol{u} = 1 \tag{9b}$$

$$u_i \geq 0 \quad \forall i \in [n] \tag{9c}$$

$$dy_i(t) = \mu_i y_i(t)dt + \sigma_i y_i(t)dW_i(t) \quad \forall i \in [n] \tag{9d}$$

$$y_i(0) = \zeta_i \quad \forall i \in [n], \tag{9e}$$

where $\boldsymbol{y}(t) \in \mathbb{R}^n$ represents the asset prices trend and $\boldsymbol{y}(T)$ denotes the asset prices at time horizon $T$ in (9a). The asset price dynamics described by (9d) follow a stochastic differential equation with drift $\mu_i$, volatility $\sigma_i$, and Wiener process $W_i(t)$ (Rudzis, 2017). Decisions $\boldsymbol{u} \in \mathbb{R}^n$ represent fractional portfolio allocations. The objective minimizes $J(\boldsymbol{u}, \boldsymbol{y}(t)) = L(\boldsymbol{u}, \boldsymbol{y}(T))$ with $\Phi(\boldsymbol{u}, \boldsymbol{y}(t), t) = 0$, balancing risk via covariance matrix $\boldsymbol{\Sigma}$ and expected return $\boldsymbol{y}(T)^\top \boldsymbol{u}$.

**Datasets and methods.** The drift and volatility factors $\mu_i \sim \mathcal{U}(0.5, 1)$ and $\sigma_i \sim \mathcal{U}(0.05, 0.1)$ in (9d) are sampled from uniform distributions. Following (Sambharya et al., 2022), initial asset prices $\{\zeta_i\}_{i=1}^n$ are obtained from the Nasdaq database (Nasdaq, 2022) to form initial vectors $\{\boldsymbol{\zeta}^j\}_{j=1}^{10,000}$, split into 80% training, 10% validation, and 10% test sets. Asset price trends $\boldsymbol{y}(t)$ are generated using an SDE solver with Itô integration (Kloeden & Platen, 2023). Given $\boldsymbol{y}(T)$, the convex solver `cvxpy` (Diamond & Boyd, 2016) computes the optimal decision $\boldsymbol{u}^\star$ for supervision during training.

We evaluate the role of asset price predictors using three models: a neural-SDE model, an LSTM, and a 2-layer Feed Forward ReLU network, each compared to a numerical SDE solver, which is discussed in Appendix E, Figure 8. Each model estimates final asset prices $\hat{\boldsymbol{y}}(T)$, which inform the DE-OP model $\mathcal{F}_\omega$ to estimate optimal decision allocations $\hat{\boldsymbol{u}} = \mathcal{F}_\omega(\hat{\boldsymbol{y}}(T))$. A "static" baseline method uses only proxy optimizers (Lagrangian Dual, DC3, or PDL) to approximate decisions based on initial prices $\boldsymbol{y}(0) = \boldsymbol{\zeta}_i$. Detailed comparisons of these approaches are provided in Appendix E.2.

**Results.** A comparison between the DE-OP and the baseline methods is provided in Figure 3. The figure reports experiments for $n = 50$ variable, while additional experiments are relegated to

the Appendix E.2. The x-axis categorizes the methods based on the type of asset price predictor used, or the lack of thereof, and reports the average *optimality gap* (in percentage) on the test set, defined as $\frac{|L(\boldsymbol{u}^\star(\boldsymbol{y}(T)),\boldsymbol{y}(T)) - L(\hat{\boldsymbol{u}}(\hat{\boldsymbol{y}}(T)),\boldsymbol{y}(T))|}{|L(\boldsymbol{u}^\star(\boldsymbol{y}(T)),\boldsymbol{y}(T))|} \times 100$. It measures the sub-optimality of the predicted solutions $\hat{\boldsymbol{u}}$ with respect to ground truth final asset price $\boldsymbol{y}(T)$, across different methods.

The figure highlights the substantial performance difference between the dynamic DE-OP models and the static-only proxy optimizer methods, marked in the last column, denoted by $\mathcal{N}_\theta = \emptyset$. These static methods (DC3, LD, and PDL) fail to incorporate asset price dynamics, resulting in notably higher optimality gaps (exceeding 100%). *Notably, their predictions can be over twice as suboptimal as the optimal solutions derived from dynamic modeling.*

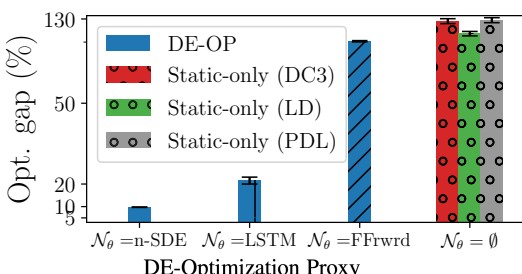

Figure 3: Average Opt. gap with $n = 50$ asset prices.

In contrast, DE-OP models that incorporate SDE models significantly outperform static methods. In particular, using neural-SDE predictors to model asset price dynamics results in *much higher* decision quality compared to both LSTM and Feed Forward models. Specifically, DE-OP with neural-SDE achieves the lowest optimality gap at 9.11%, successfully capturing the dynamics through an explicit modeling of the asset prices' governing equations. In contrast, the LSTM model results in a notably higher optimality gap of 21.17%, approximately 2.5 times greater than that of DE-OP with neural-SDE, attributable to its lack of dynamic modeling. The Feed Forward model performs significantly worse with an optimality gap of 102.45% for LD, indicating its inability to capture the time-dependent nature of asset pricing data.

The dynamic forecasting results in Appendix E.1 display different levels of precision of the final asset prices predictions among the dynamic predictors considered, which ultimately led to different decision quality. *In particular, DE-OP with a neural-SDE model performs consistently better than the LSTM model and produces up to $25\times$ better decisions (measured in terms of optimality gap) than any static-only proxy optimizer method.* This stark contrast underscores the effectiveness of DE-OP models in leveraging dynamic asset price predictors to improve decision quality.

## 5.2 STABILITY-CONSTRAINED AC-OPTIMAL POWER FLOW

We now turn on studying a real-world problem in power systems, which arises from integration of Synchronous Generator Dynamics with the Alternating Current Optimal Power Flow (AC OPF) problem. The AC OPF, detailed in Appendix Model (1), is foundational in power systems for finding cost-effective generator dispatches that meet demand while complying with physical and engineering constraints. Traditionally addressed as a *steady-state* snapshot, the AC OPF problem requires frequent resolution (e.g., every 10-15 minutes) due to fluctuating loads, posing challenges in maintaining operational continuity and system stability (Hatziargyriou et al., 2021). Given the non-convexity, high dimensionality, and computational demands of this problem, proxy optimizers have emerged as a viable alternative to traditional numerical solvers. However, as shown later in this section, existing approaches such as those in Donti et al. (2020) and Fioretto et al. (2020), which focus on the steady-state aspect, fail to address the dynamic system requirements adequately.

The integration of the generator dynamics and related stability constraints into the steady-state AC-OPF formulation leads to the *stability-constrained* AC-OPF problem, which is detailed in Appendix B.3. Here, the decision variables $\boldsymbol{u}$, comprising generator power outputs and bus voltages, influence the state variables $\boldsymbol{y}(t)$, representing generator rotor angles and speeds. This coupling renders the problem particularly challenging. The objective is to optimize power dispatch costs, while satisfying demand, network constraints, and ensuring system stability as described by (1d). DE-OP enables, for the first time to our knowledge, the integration of generator dynamics within the optimization process. This integration is key for system stability. Our implementation uses neural-ODE (Chen et al., 2018) models $\mathcal{N}_\theta$, assigned individually to each generator. A comparative analysis of neural-ODEs, PINNs and a numerical solver for capturing these dynamics is available in Appendix D.1.

**Datasets.** DE-OP is evaluated on two key power networks, the WSCC 9 and IEEE 57 bus-systems (Babaeinejadsarookolaee et al., 2021), under various operating system conditions, as studied in

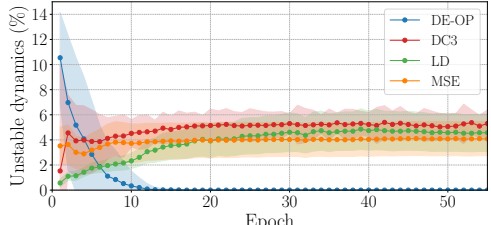 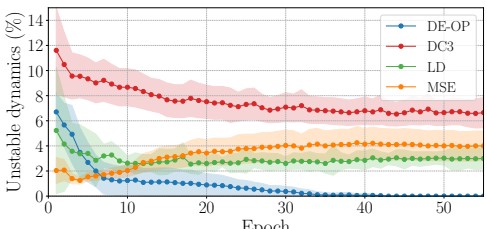

Figure 4: WSCC 9-bus system - Percentage of unstable dynamics at training time for different methods based on 40 trials.

Figure 5: IEEE 57-bus system - Percentage of unstable dynamics at training time for different methods based on 40 trials.

| Models | | Metrics | | | |
|---|---|---|---|---|---|
| $\mathcal{F}_\omega$ | $\mathcal{N}_\theta$ | Stability Vio. | Flow Vio. $\times 10^{-3}$ | Boundary Vio. $\times 10^{-4}$ | Optimality gap* (at steady-state) % |
| DE-OP (ours) | | **0.00** | $9.15 \pm 0.442$ | $0.25 \pm 0.172$ | $0.22 \pm 0.02$ |
| MSE | $\emptyset$ | $23.30 \pm 0.206$ | $12.65 \pm 2.281$ | $6.44 \pm 1.434$ | $0.17 \pm 0.02$ |
| LD | $\emptyset$ | $23.10 \pm 0.219$ | $6.23 \pm 0.125$ | $0.00$ | $0.17 \pm 0.01$ |
| DC3 | $\emptyset$ | $28.60 \pm 0.232$ | $0.00$ | $0.00$ | $0.16 \pm 0.01$ |

Table 1: Average and standard deviation of constraint violations and (steady-state) optimality gap on the IEEE 57-bus system for different approaches based on 40 independent runs.

(Fioretto et al., 2020). This assessment benchmarks our proposed method, DE-OP, against three leading proxy optimizer methods for AC-OPF, which operate under a "steady-state" assumption, and thus cannot cope with constraints (1b) and (1c) of the DE-constrained problem. These methods are LD (Fioretto et al., 2020), DC3 (Donti et al., 2020), and MSE Zamzam & Baker (2020), introduced in details in Appendix C.1.

Furthermore, discretizing the DE system through methods like direct collocation (Betts, 2010), or iteratively optimizing over the time horizon adopting Model Predictive Control-based techniques, becomes highly impractical for real-time applications due to the high number of variables and non-linear system dynamics associated with each generator in the system. In contrast, all methods used for comparison (including our DE-OP) *produce estimates of the optimal decision variables within milliseconds*, as shown in Appendix, Table 6. Additionally, we note that only fast inference times are of interest in the area of proxy optimizers, as once trained, these methods can be applied to various related but distinct problem instances. Crucially, as the generator dynamics are captured by separate neural-DE models, their computation is fully parallelized and suggests potential for DE-OP in large-scale networks, where proxy optimizers have shown promising results (Mak et al., 2023).

For both the benchmark systems, DE-OP and each proxy optimizer model are trained on a dataset $\mathcal{D} = \{(\boldsymbol{\zeta}_i, \boldsymbol{u}_i^\star)\}_{i=1}^{10,000}$, where $\boldsymbol{\zeta}_i$ representing a load demand and $\boldsymbol{u}_i^\star$ the corresponding optimal, *steady-state* decision. The $j^{th}$ load of the $i$-th sample $\boldsymbol{\zeta}_i^j$ is generated by applying a uniform random perturbation of $\pm 20\%$ to the corresponding nominal load. The (steady-state) AC-OPF models are implemented in Julia and solved using IPOPT (Wächter & Biegler, 2006). The dataset uses an 80/10/10 split. By leveraging the knowledge of the decision variables' bounds (see Appendix, Model 2, Constraints (11b)-(11c)), each neural-ODE model $\mathcal{N}_\theta$ is trained to learn the corresponding generator dynamics on a dataset of near-optimal decisions $\Pi_{u'}$ as described in Section 4.2. We refer the reader to Appendix B.3 for further details on the neural-ODE models training.

**Results.** Figures 4 and 5 show the percentage of estimated decisions violating the stability constraints during the first 50 epochs of training, across all methods and test cases. On the WSCC 9-bus system (Figure 4) DE-OP learns rapidly to meet the dynamic constraints, which violations approaches zero level after epoch 10 of training, whereas, all the baseline methods lacking dynamic modeling, consistently produce unstable dynamics. Notably, all the baseline methods tested, e.g., DC3, LD, and MSE, systematically fail to satisfy stability requirements. In contrast, by integrating

generator dynamics within its training model, DE-OP begins to satisfy these requirements early in training, as depicted in both figures (blue curves). DE-OP shows a rapid adjustment on both systems within the first few epochs, and bringing the violations to near zero. In contrast, all baseline methods continue to exhibit 4% to 8% unstable dynamics throughout the training, even for a much higher number of training epochs. As we will show later, these will reflect also in large stability constraints violations, when evaluated on the test set.

Table 1 displays the test-set results for DE-OP and the baseline methods on the IEEE 57-bus system. For a detailed discussion on the results of each method on the WSCC-9 bus system, please see Appendix D and Table 5. These tables reports the following metrics:

- *Static and Stability Constraint Violations*: These are quantified for each test instance. The $j$-th static equality and the $k$-th inequality violation are calculated as $\frac{1}{n_{\text{test}}} \sum_{i=1}^{n_{\text{test}}} |h'_j(\hat{\boldsymbol{u}}^i, \hat{\boldsymbol{y}}^i(t))|$ and $\frac{1}{n_{\text{test}}} \sum_{i=1}^{n_{\text{test}}} \max(0, g_k(\hat{\boldsymbol{u}}^i, \hat{\boldsymbol{y}}^i(t)))$ respectively, where $n_{\text{test}}$ is the test-set size. Detailed descriptions of the problem constraints can be found in Appendices B.1 and B.3.
- *Optimality gap (at steady-state)*: This metric is defined as $\frac{|L(\boldsymbol{u}^\star(\zeta), \boldsymbol{y}^\star(T)) - L(\hat{\boldsymbol{u}}(\zeta), \hat{\boldsymbol{y}}(T))|}{|L(\boldsymbol{u}^\star(\zeta), \boldsymbol{y}^\star(T))|} \times 100$. It measures the gap incurred by the predictions $\hat{\boldsymbol{u}}, \hat{\boldsymbol{y}}(t)$ against the decisions $\boldsymbol{u}^\star$ which are computed under the assumption that the generators are in a steady-state condition. This assumption is crucial for evaluating how closely each solution approximates the AC-OPF optimal results, though it *does not necessarily reflect the results relative to the stability-constrained AC-OPF problem*, our main focus, but which is highly intractable. Given the non-linearity of both the dynamics and optimization in the stability-constrained AC-OPF, computing exact optimal decisions $\boldsymbol{u}^\star$ with traditional methods is not feasible. Consequently, while our method may show slightly higher steady-state optimality gaps, these should not be interpreted in the context of the dynamic problem.

Firstly, note that all methods report comparable static constraint violations and steady-state optimality gaps, with errors within the $10^{-3}$ to $10^{-4}$ range. Although DE-OP exhibits slightly higher steady-state optimality gaps, approximately 0.05% higher than the best performing baseline, it's important to recall that this metric *does not* reflect the stability-constrained AC-OPF optimality gap, but rather that of the problem addressed by the baselines, placing DE-OP at a seeming disadvantage. The higher objective costs observed with DE-OP is intuitively attributed to a restricted feasible space resulting from the integration of generator stability constraints within the AC OPF problem.

Crucially, all baseline methods systematically fail to meet the stability requirements, often by margins exceeding those observed during training. This illustrates a typical scenario where prediction errors on decisions parametrizing system dynamics have cascading effects on the associated constraint violations. In stark contrast, DE-OP achieves full compliance with stability constraints, reporting zero violations in each instance analyzed.

These findings are important. They highlight the critical role of dynamic requirements in AC-OPF problems for achieving accurate and stable solutions. The results underscore DE-OP's effectiveness in adjusting potentially unstable set points, as further detailed in Appendix D.2 and demonstrates DE-OP's effectiveness in ensuring system stability compared to traditional methods that focus on optimality under steady-state assumptions.

## 6 CONCLUSION

This work was motivated by the efficiency requirements associated with solving differential equations (DE)-constrained optimization problems. It introduced a novel learning-based framework, DE-OP, which incorporate differential equation constraints into optimization tasks for near real-time application. The approach uses a dual-network architecture, with one approximating the control strategies, focusing on steady-state constraints, and another solving the associated DEs. This architecture exploits a primal-dual method to ensure that both the dynamics dictated by the DEs and the optimization objectives are concurrently learned and respected. This integration allows for end-to-end differentiation enabling efficient gradient-based optimization, and, for the first time to our knowledge, solving DE-constrained optimization problems in near real-time. Empirical evaluations across financial modeling and energy optimization tasks, illustrated DE-OP's capability to adeptly address these complex challenges. The results demonstrate not only the effectiveness of our approach but also its broad potential applicability across various scientific and engineering domains where system dynamics are crucial to optimization or control processes.

## 7 ACKNOWLEDGMENTS

This work was partially supported by NSF grants EPCN-2242931 and NSF CAREER-2143706. The view and conclusions are those of the authors only.

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

# A Stochastic Differential Equation Constrained Optimization

This section extends the problem description (1) presented in Section 3 of the main paper to a stochastic setting. In presence of stochastic dynamics, the optimization problem constrained by differential equations (1) becomes

$$\underset{\boldsymbol{u}}{\text{Minimize}} \ \ \mathbb{E}\left[L(\boldsymbol{u}, \boldsymbol{y}(T)) + \int_{t=0}^{T} \Phi(\boldsymbol{u}, \boldsymbol{y}(t), t)\, dt\right] \tag{10a}$$

$$\text{s.t.} \ \ d\boldsymbol{y}(t) = \boldsymbol{F}(\boldsymbol{u}, \boldsymbol{y}(t), t)dt + \boldsymbol{G}(\boldsymbol{u}, \boldsymbol{y}(t), t)d\boldsymbol{W}(t) \tag{10b}$$

$$\boldsymbol{y}(0) = \boldsymbol{I}(\boldsymbol{u}) \tag{10c}$$

$$\boldsymbol{g}(\boldsymbol{u}, \boldsymbol{y}(t)) \leq 0 \tag{10d}$$

$$\boldsymbol{h}(\boldsymbol{u}, \boldsymbol{y}(t)) = 0. \tag{10e}$$

The SDE constrained optimization problem (10) involves determining the optimal decision variables $\boldsymbol{u} = (u_1, \ldots, u_n)$ in a system where the state variables $\boldsymbol{y}(t) = (y_1(t), \ldots, y_m(t))$ evolve according to stochastic dynamics (10a) and initial conditions dictated by (10b). Each state variable $y_i(t)$ is governed by a stochastic differential equation $d\boldsymbol{y}_i(t) = F_i(\boldsymbol{y}(t), \boldsymbol{u}, t)dt + G_i(\boldsymbol{y}(t), \boldsymbol{u}, t)dW_i(t)$, where $F_i$ represents the deterministic part of the dynamics, and $G_i$ captures the stochastic component, where $W_i(t)$ is a Wiener process. The set of all such equations is described by $\boldsymbol{F}$ and $\boldsymbol{G}$, as defined by (10b). The initial condition for the state variables is set by constraints (10c), where $\boldsymbol{y}(0) = \boldsymbol{I}(\boldsymbol{u})$ defines the starting state based on the control variables $\boldsymbol{u}$. Constraints (10d) and (10e) enforce inequality and equality constraints, respectively, on the state and control variables, ensuring that the system behaves within specified bounds throughout the decision process.

The objective (10a) is to minimize the expected value of a combination of the running cost $\Phi(\boldsymbol{u}, \boldsymbol{y}(t), t)$, which varies with the state and decision variables over time, and the terminal cost $L(\boldsymbol{u}, \boldsymbol{y}(T))$, which depends on the final state $\boldsymbol{y}(T)$ and the decision variables $\boldsymbol{u}$. The optimization is performed over a time horizon $T$, which defines the period during which the decision-making process occurs.

# B Stability Constrained AC-Optimal Power Flow

This section describes the stability constrained AC-Optimal Power Flow problem; it first introduces the AC-Optimal Power Flow problem and the synchronous generator dynamics, to eventually integrate these two components to form the stability constrained AC-Optimal Power Flow problem.

## B.1 AC-Optimal Power Flow problem

The AC-Optimal Power Flow (OPF) problem determines the most cost-effective generator dispatch that satisfies demand within a power network subject to various physical and engineering power systems constraints. Typically, the OPF problem involves capturing a snapshot of the power network parameters and determine the bus voltages and generator set-points based on that fixed state. A power network can be represented as a graph $(\mathcal{N}, \mathcal{L})$ with the node set $\mathcal{N}$ consisting of $n$ buses, and the edge set $\mathcal{L}$ comprises $l$ lines. The set $\mathcal{L}$ is defined as a collection of directed arcs, with $\mathcal{L}^R$ indicating the arcs in $\mathcal{L}$ but in the opposite direction. $\mathcal{G} \subset \mathcal{N}$ represents the set of all synchronous generators in the system. The power generation and demand at a bus $i \in \mathcal{N}$ are represented by complex variables $S_i^r = p_i^r + jq_i^r$ and $S_i^d = p_i^d + jq_i^d$, respectively. The power flow across line $ij$ is denoted by $S_{ij}$, and $\theta_i$ symbolizes the phase angles at bus $i \in \mathcal{N}$.

The AC power flow equations use complex numbers for current $I$, voltage $V$, admittance $Y$, and power $S$, interconnected through various constraints. Kirchhoff's Current Law (KCL) is represented by $I_i^r - I_i^d = \sum_{(i,j) \in \mathcal{L} \cup \mathcal{L}^R} I_{ij}$, Ohm's Law by $I_{ij} = Y_{ij}(V_i - V_j)$, and AC power flow is denoted as $S_{ij} = V_i I_{ij}^*$. These principles form the AC Power Flow equations, described by (11f) and (11g), which formulation is described by Model 1. The goal is to minimize a function (11a) representing dispatch costs for each generator. Constraints (11b)-(11c) represents voltage operational limits to bound voltage magnitudes and phase angle differences, while (11d)-(11e) set boundaries for generator output and line flow. Constraint (11h) sets the reference phase angle. Finally, constraints (11f) and (11g) enforce KCL and Ohm's Law, respectively. The classical, *steady-state* problem,

---

**Model 1** The AC Optimal Power Flow Problem (AC-OPF)

$$\textbf{Parameters}: \quad \boldsymbol{\zeta} = (\boldsymbol{S}^d)$$

$$\textbf{decision variables}: \quad \boldsymbol{u} = (S_i^r, V_i) \;\; \forall i \in \mathcal{N}, \;\; S_{ij} \;\; \forall (i,j) \in \mathcal{L}$$

$$\text{Minimize} \sum_{i \in \mathcal{G}} c_{2i}(\Re(S_i^r))^2 + c_{1i}\Re(S_i^r) + c_{0i} \tag{11a}$$

s. t.

$$v_i^l \leq |V_i| \leq v_i^u \;\; \forall i \in N \tag{11b}$$

$$-\theta_{ij}^\Delta \leq \angle(V_iV_j^*) \leq \theta_{ij}^\Delta \;\; \forall (i,j) \in \mathcal{L} \tag{11c}$$

$$S_i^{rl} \leq S_i^r \leq S_i^{ru} \;\; \forall i \in \mathcal{N} \tag{11d}$$

$$|S_{ij}| \leq s_{ij}^u \;\; \forall (i,j) \in \mathcal{L} \tag{11e}$$

$$S_i^r - S_i^d = \sum_{(i,j) \in L} S_{ij} \;\; \forall i \in \mathcal{N} \tag{11f}$$

$$S_{ij} = Y_{ij}^*|V_i|^2 - Y_{ij}^*V_iV_j^* \;\; \forall (i,j) \in \mathcal{L} \tag{11g}$$

$$\theta_{\text{ref}} = 0 \tag{11h}$$

---

described by Model 1, does *not* incorporate systems dynamics capturing the behavior of the synchronous generators, and as such, does *not* guarantee stable operations for a power system. This paper extends this problem by introducing the *Stability-Constrained* AC-Optimal Power Flow Problem, which integrates the generator dynamics and related stability constraints within the AC-OPF problem (1).

### B.2 GENERATOR DYNAMICS

The generator dynamics are modeled using the "Classical machine model" (12), which is typically adopted to describe the dynamic behavior of synchronous generators (Sauer & Pai, 1998)

$$\frac{d}{dt} \begin{bmatrix} \delta^g(t) \\ \omega^g(t) \end{bmatrix} = \begin{bmatrix} \omega_s(\omega^g(t) - \omega_s) \\ \frac{1}{M^g}\left(P_m^g - D^g(\omega^g(t) - \omega_s) - \frac{E_{q,0}^g v_g}{X_d^g}\sin(\delta^g(t) - \theta_g)\right) \end{bmatrix} \tag{12}$$

, where $\delta^g(t)$ and $\omega^g(t)$ represents the rotor angle and angular speed over time $t$ of generator $g \in \mathcal{G}$, $\omega_s$ the synchronous angular frequency, $M^g$ the machine's inertia constant, $D^g$ the damping coefficient, $P_m^g$ the mechanical power, $X_d^g$ the transient reactance and $E_{q,0}^g$ electromotive force. The initial value of the rotor angle $\delta_0^g$, and electromotive force $E_{q,0}^g$ for each generator $g \in \mathcal{G}$ are derived from the active and reactive power equations, assuming the generator dynamical system (12) being in a steady state condition at time instant $t = 0$, $\frac{d}{dt}[\delta^g(t)\,\omega^g(t)]_{t=0}^T = [0\,0]^T$:

$$\frac{E_{q,0}^g v_g \sin(\delta_0^g - \theta_g)}{X_d^g} - p_g^r = 0, \tag{13}$$

$$\frac{E_{q,0}^g v_g \cos(\delta_0^g - \theta_g) - v_g^2}{X_d^g} - q_g^r = 0. \tag{14}$$

Following the same assumptions, the initial rotor angular speed is set as

$$\omega_0^g = \omega_s. \tag{15}$$

**Stability limit** To guarantee stability of a synchronous generator $g \in \mathcal{G}$, the rotor angle $\delta^g(t)$ is required to remain below an instability threshold $\delta^{\max}$, as defined by SIngle Machine Equivalent (SIME) model:

$$\delta^g(t) \leq \delta^{\max} \quad \forall t \geq 0. \tag{17}$$

Unstable conditions arise when violating the inequality constraint (17), which is the principal binding constraint that necessitates re-dispatching.

---

**Model 2** The Stability Constrained AC-OPF Problem

$$\textbf{Parameters}: \quad \boldsymbol{\zeta} = (\boldsymbol{S}^d)$$

$$\textbf{decision variables}: \quad \boldsymbol{u} = (S_i^r, V_i) \;\; \forall i \in \mathcal{N}, \;\; S_{ij} \;\; \forall (i,j) \in \mathcal{L}$$

$$\textbf{State variables}: \quad \boldsymbol{y}(t) = (\boldsymbol{\delta}^g(t), \boldsymbol{\omega}^g(t)) \;\; \forall g \in \mathcal{G}$$

$$\text{Minimize} \quad \sum_{i \in \mathcal{G}} c_{2i}(\Re(S_i^r))^2 + c_{1i}\Re(S_i^r) + c_{0i} \tag{16a}$$

s. t.

$$(11b) - (11h) \tag{16b}$$

$$\frac{d\delta^g(t)}{dt} = \omega_s(\omega^g(t) - \omega_s) \;\; \forall g \in \mathcal{G} \tag{16c}$$

$$\frac{d\omega^g(t)}{dt} = \frac{1}{m^g}\left(p_m^g - d^g(\omega^g(t) - \omega_s)\right)$$
$$\qquad - \frac{e_q'^g(0)|V_g|}{x_d'^g m^g}\sin(\delta^g(t) - \theta_g) \;\; \forall g \in \mathcal{G} \tag{16d}$$

$$\frac{e_q'^g(0)|V_g|\sin(\delta^g(0) - \theta_g)}{x_d'^g} - p_g^r = 0 \;\; \forall g \in \mathcal{G} \tag{16e}$$

$$\frac{e_q'^g(0)|V_g|\cos(\delta^g(0) - \theta_g) - |V_g|^2}{x_d'^g} - q_g^r = 0 \;\; \forall g \in \mathcal{G} \tag{16f}$$

$$\omega^g(0) = \omega_s \;\; \forall g \in \mathcal{G} \tag{16g}$$

$$\delta^g(t) \leq \delta^{\max} \;\; \forall g \in \mathcal{G}. \tag{16h}$$

$$\tag{16i}$$

---

## B.3 STABILITY-CONSTRAINED AC-OPTIMAL POWER FLOW PROBLEM

The generator dynamics (12) and its initial conditions equations (13)-(15), together with the associated stability constraints (17), are thus integrated within the steady-state AC-OPF Problem 1, giving rise to the Stability-Constrained AC-OPF problem, which is detailed in Model 2. In this problem, parameters $\boldsymbol{\zeta} = \boldsymbol{S}^d$ represent customer demand, while decision variables $\boldsymbol{x} = (\boldsymbol{S}^r, \boldsymbol{V})$ are the generator settings and bus voltages; the state variables $\boldsymbol{y}(t) = (\delta^g(t), \omega^g(t)) \; \forall g \in \mathcal{G}$ represent the rotor angle and angular speed of the generators.

**Training setting of Neural Ordinary Differential Equation Models** As the generator dynamics are described by a system of ODEs, neural-ODE (Chen et al., 2018) models, one for each synchronous generator $g \in \mathcal{G}$, are used to capture their dynamics. Each neural-DE model $\mathcal{N}_\theta^g$ is trained in a supervised fashion, as described in Section 4.2, to obtain dynamic predictors that are capable of providing accurate estimate of the state variables $\boldsymbol{y}^g(t)$ across a family of instances of the generator model (12). Specifically, for each generator $g \in \mathcal{G}$, the datasets $\mathcal{D}^g$ used for training the generator dynamic predictor $\mathcal{N}_\theta^g$, consists of pairs $(\boldsymbol{x}_i, \boldsymbol{y}_i(t)) \sim \mathcal{D}^g$, where $\boldsymbol{x}_i = (\delta_0^g, \omega_0^g, |V_g'|, \theta_g')$ is the input of the neural-ODE model, and $\boldsymbol{y}_i(t) = (\delta^g(t), \omega^g(t), |V_g'|(t), \theta_g'(t))$ the corresponding solution of (12) with initial conditions $\boldsymbol{y}_i(0) = \boldsymbol{I}(\boldsymbol{x}_i)$, represented by (16e)-(16g) and computed using $Dopri5$, a numerical algorithm implementing an adaptive Runge-Kutta method. For each input $\boldsymbol{x}$, the OPF decision variables $|V_g'|, \theta_g'$ are sampled from a uniform distribution $\mathcal{U}(a,b)$, where $a$ and $b$ are given by the corresponding operational limits specified by Constraints (11b)-(11c). Note that each of these variables influences the initial condition of the state variables $(\delta^g(0), \omega^g(0))$ via Equation (13)-(15), as well as the governing equations of the generator. As $|V_g'|, \theta_g'$ extend the actual generator state variables $(\delta^g(t), \omega^g(t))$, we are implicitly augmenting the generator model with these two additional state variables that have no dynamics (e.g. $\frac{d|V_g'|(t)}{dt} = 0, \frac{d\theta_g(t)}{dt} = 0$) and initial condition $|V_g'|(0) = |V_g'|, \theta_g'(0) = \theta_g'$. This trick allows us to explicitly inform the neural ODE model of the role played by the voltage magnitude $|V_g|$ and angle $\theta_g$ on the dynamics of each generator. The generator characteristics parameters of model (12), such as the damping coefficient $D^g$, inertia constant $M^g$, and mechanical power $P_m^g$ are adopted from Li et al. (2016). Each dataset $\mathcal{D}^g$ contains approximately $50\%$ of unstable trajectories and $50\%$ of stable trajectories, generated as described in Section 4.2. At training time, given a pair $(\boldsymbol{x}_i, \boldsymbol{y}_i(t)) \sim \mathcal{D}^g$, the target is constructed as

$\boldsymbol{y}_i^g(t) = \boldsymbol{y}_i^g(0), \boldsymbol{y}_i^g(\Delta_t), \ldots, \boldsymbol{y}_i^g(n\Delta_t)$, with $\Delta_t = 0.001$ and the number of points $n$, is set to 200 at the beginning of the training, and gradually increases up to 1000. This trick allows to avoid local minima during training (Kidger, 2022). At test time, we set $n = 1000$.

### B.4 DYNAMIC PORTFOLIO OPTIMIZATION

DE-OP uses a neural-SDE (Kidger et al., 2021) model $\mathcal{N}_\theta$ to capture the asset price dynamics $\boldsymbol{y}(t)$. The neural-SDE model consists of 2 separate neural network, $\mathcal{N}_\theta = (\mathcal{N}_\theta^f, \mathcal{N}_\theta^g)$ where $\mathcal{N}_\theta^f$ aims to capture the deterministic component of (9d), $\mu_i \boldsymbol{\zeta}(t)dt$ and $\mathcal{N}_\theta^g$ the stochastic component $\sigma_i \boldsymbol{\zeta}(t)dW_i(t)$. $\mathcal{N}_\theta^f$ is a simple linear layer, while $\mathcal{N}_\theta^g$ is a 2-layer ReLU neural network. Given the initial asset price vector $\boldsymbol{y}(0) = \boldsymbol{\zeta}$, the neural-SDE model $\mathcal{N}_\theta$ generates an estimate $\hat{\boldsymbol{y}}(t) = \mathcal{N}_\theta(\boldsymbol{y}(0), t)$ of the asset prices trend, from which the final asset price $\hat{\boldsymbol{y}}(T)$ is obtained. The LSTM and the Feed Forward model used to estimate the final asset price $\hat{\boldsymbol{y}}(T)$ as the dynamic component of the corresponding baseline method are both a 2-layer ReLU neural network. The final time instant $T = 28,800$ seconds which corresponds to 8 hours. Given initial condition $\boldsymbol{y}^j(0) = \boldsymbol{\zeta}^j$, the asset price trend $\boldsymbol{y}^j(t)$ is obtained by Ito numerical integration of (9d). The neural-SDE model is trained on a dataset $\{(\boldsymbol{\zeta}^j, \boldsymbol{y}^j(t))\}_{j=1}^N$; the LSTM model is trained on a dataset $\{(\boldsymbol{y}^j(t), \boldsymbol{y}^j(T))\}_{j=1}^N$, where $\boldsymbol{y}^j(t) = \boldsymbol{y}^j(0), \boldsymbol{y}^j(\Delta_t), \ldots, \boldsymbol{y}^j(\Delta_t K)$ is a time series, $\Delta_t = 100$ seconds and $\Delta_t K = T-1$. The Feed Forward network is trained on a dataset $\{(\boldsymbol{y}^j(0), \boldsymbol{y}^j(T))\}_{j=1}^N$.

## C ADDITIONAL EXPERIMENTAL DETAILS

### C.1 PROXY OPTIMIZER METHODS

This subsection describes in brief the proxy optimizer methods adopted in the experiments to estimate the optimal decision variables $\boldsymbol{u}^\star$, within the operational setting described by (1) or (10). Each description below assumes a DNN model $\mathcal{F}_\omega$ parameterized by $\omega$, which acts on problem parameters $\boldsymbol{\zeta}$ to produce an estimate of the decision variables $\hat{\boldsymbol{u}} := \mathcal{F}_\omega(\boldsymbol{\zeta})$, so that $\hat{\boldsymbol{u}} \approx \boldsymbol{u}^\star(\boldsymbol{\zeta})$. To ease notation, the dependency from problem parameters $\boldsymbol{\zeta}$ is omitted.

**Lagrangian Dual Learning (LD).** Fioretto et al. (2020) constructs the following modified Lagrangian as a loss function:

$$\mathcal{L}^{LD}(\hat{\boldsymbol{u}}, \boldsymbol{y}(t)) = \|\hat{\boldsymbol{u}} - \boldsymbol{u}^\star(\boldsymbol{\zeta})\|^2 + \boldsymbol{\lambda}^T \left[\boldsymbol{g}(\hat{\boldsymbol{u}}, \boldsymbol{y}(t))\right]_+ + \boldsymbol{\mu}^T \boldsymbol{h}(\hat{\boldsymbol{u}}, \boldsymbol{y}(t)).$$

At each iteration of LD training, the model $\mathcal{F}_\omega$ is trained to minimize a balance of constraint violations and proximity to the precomputed target optima $\boldsymbol{u}^\star(\boldsymbol{\zeta})$. Updates to the multiplier vectors $\boldsymbol{\lambda}$ and $\boldsymbol{\mu}$ are calculated based on the average constraint violations incurred by the predictions $\hat{\boldsymbol{u}}$, mimicking a dual ascent method (Boyd et al., 2011).

**Deep Constraint Completion and Correction (DC3).** Donti et al. (2020) uses the loss function

$$\mathcal{L}^{\text{DC3}}(\hat{\boldsymbol{u}}, \hat{\boldsymbol{y}}(t)) = \mathcal{J}(\hat{\boldsymbol{u}}, \hat{\boldsymbol{y}}(t)) + \lambda\| \left[\boldsymbol{g}(\hat{\boldsymbol{u}}, \hat{\boldsymbol{y}}(t))\right]_+ \|^2 + \mu\|\boldsymbol{h}(\hat{\boldsymbol{u}}, \hat{\boldsymbol{y}}(t))\|^2$$

which relies on a completion-correction technique to enforce constraint satisfaction, while maximizing the empirical objective $\mathcal{J}$ in self-supervised training.

**Self-Supervised Primal-Dual Learning (PDL).** Park & Van Hentenryck (2023) uses an augmented Lagrangian loss function

$$\mathcal{L}^{\text{PDL}}(\hat{\boldsymbol{u}}, \hat{\boldsymbol{y}}(t)) = \mathcal{J}(\hat{\boldsymbol{u}}, \boldsymbol{y}(t)) + \hat{\boldsymbol{\lambda}}^T \boldsymbol{g}(\hat{\boldsymbol{u}}, \hat{\boldsymbol{y}}(t)) + \hat{\boldsymbol{\mu}}^T \boldsymbol{h}(\hat{\boldsymbol{u}}, \hat{\boldsymbol{y}}(t)) +$$

$$\frac{\rho}{2} \left( \sum_j [g_j(\hat{\boldsymbol{u}}, \hat{\boldsymbol{y}}(t))]_+ + \sum_i |h_i(\hat{\boldsymbol{u}}, \hat{\boldsymbol{y}}(t))| \right),$$

which consists of a primal network to approximate the decision variables, and a dual network to learn the Lagrangian multipliers update. The method is self-supervised, requiring no precomputation of target solutions for training.

**MSE.** Zamzam & Baker (2020) uses the loss function:

$$\mathcal{L}^{\text{MSE}}(\hat{\boldsymbol{u}}, \boldsymbol{u}^{\star}) = \|\hat{\boldsymbol{u}} - \boldsymbol{u}^{\star}\|^2$$

which minimizes the MSE between the predicted solution $\hat{\boldsymbol{u}}$ and the corresponding precomputed solution $\boldsymbol{u}^{\star}$.

### C.2  STABILITY CONSTRAINED AC-OPTIMAL POWER FLOW EXPERIMENT

**Hyperparameters of the neural-ODE models**  Each neural-ODE model is a fully connected feed-forward ReLU neural network with 2 hidden layers, each with 200 units. Each model is trained using Adam optimizer, with default learning rate $\eta = 10^{-3}$ and default hyperparameters.

**Hyperparameters of the DE-OP's optimization model and the proxy optimizer methods**  The DE-OP optimization component $\mathcal{F}_\omega$ and each baseline proxy optimizer model is trained with Adam optimizer and default learning rate $\eta = 10^{-3}$. Each proxy optimizer model is a fully connected FeedFoward ReLU neural network with 5 hidden layers, each with 200 units. The DE-OP's optimization model $\mathcal{F}_\omega$ and Lagrangian Dual proxy model are trained with a Lagrangian step size $\rho = 10^{-1}$, while the Lagrangian multipliers $\boldsymbol{\lambda}_{h'}$ and $\boldsymbol{\lambda}_g$ are updated at each epoch. DC3's proxy optimizer model is trained with the same set of hyperparameters for OPF, as reported in the original paper.

### C.3  DYNAMIC PORTFOLIO EXPERIMENT

**Hyperparameters of the asset prices predictor models**  The stochastic component of the neural-SDE, the LSTM and Feed Forward model are each 2-layers ReLU networks, each with 100 units. The neural-SDE, LSTM and Feed Forward models are all trained using Adam optimizer, with default learning rate $\eta = 10^{-3}$ and hyperparameters.

**Hyperparameters of the DE-OP's optimization model and the proxy optimizer methods**  The DE-OP optimization component $\mathcal{F}_\omega$ and each baseline proxy optimizer model is trained with Adam optimizer and default learning rate $\eta = 10^{-3}$. Each proxy optimizer model is a fully connected Feed Foward ReLU neural network with 2 hidden layers, each with 50 units. The DE-OP's optimization model $\mathcal{F}_\omega$ and Lagrangian Dual proxy model are trained with a Lagrangian step size $\rho = 10^{-1}$, while the Lagrangian multipliers $\boldsymbol{\lambda}_{h'}$ and $\boldsymbol{\lambda}_g$ are updated each 10 epoch. PDL's and DC3's proxy optimizer model uses the same hyperparameters for the Convex Quadratic task, as reported in the respective original paper.

## D  ADDITIONAL EXPERIMENTAL RESULTS: STABILITY CONSTRAINED AC OPTIMAL POWER FLOW

This section reports additional experimental results of DE-OP model and the baseline methods on the WSCC 9-bus system and the IEEE 57-bus system. Specifically, we report:

- The inference time (measured in seconds) of neural-ODE which we compare to the computational time of a traditional numerical ODE solver and the precision of state variables' estimate $\hat{\boldsymbol{y}}(t)$ (measured as the percentage $\ell^2$ error) of NODEs and PINNs Raissi et al. (2019), a different learning-based approach for learning the system dynamics.

- The (steady-state) decision error (MSE) of the OPF variables of DE-OP and each proxy optimizer method, incurred by the respective approximation $\hat{\boldsymbol{u}}$, computed assuming that the generators are in steady-state.

- The (steady-state) optimality gaps incurred by DE-OP and the baselines proxy optimizers predictions' $\hat{\boldsymbol{u}}$, and measured as $\frac{|L(\boldsymbol{u}^{\star}(\zeta), \boldsymbol{y}^{\star}(T)) - L(\hat{\boldsymbol{u}}(\zeta), \hat{\boldsymbol{y}}(T))|}{|L(\boldsymbol{u}^{\star}(\zeta), \boldsymbol{y}^{\star}(T))|} \times 100$, where $L$ is objective function (equation 16a).

- The inference time (measured in seconds) of DE-OP and each proxy optimizer model to generate $\hat{\boldsymbol{u}}$.

Table 2: Average and standard deviation of computational times for numerical solvers vs. neural-ODE inference time by method

| Method | Numerical Solver | neural-ODE |
|---|---|---|
| Dopri5 (default) | $0.135 \pm 0.015$ (sec) | $\mathbf{0.008 \pm 0}$ (sec) |
| Bosh3 | $0.446 \pm 0.039$ (sec) | $\mathbf{0.017 \pm 0}$ (sec) |

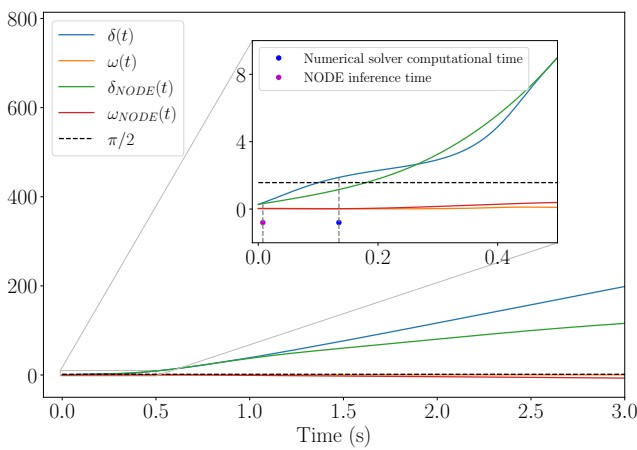

Figure 6: True and neural-ODE (NODE) solutions of the generator state variables in unstable conditions.

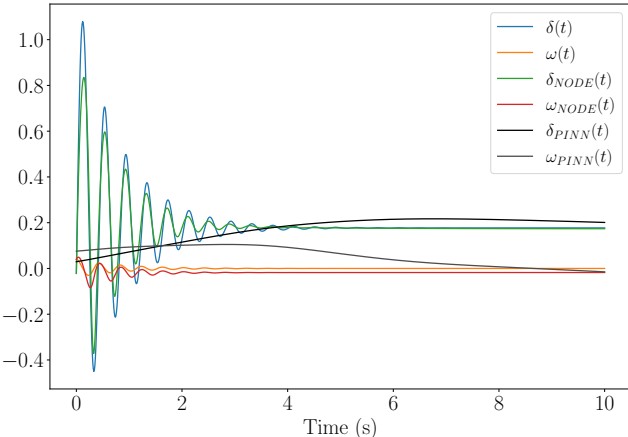

Figure 7: True, neural-ODE (NODE) and PINN estimate of the generator state variables in stable conditions.

### D.1 LEARNING THE GENERATOR DYNAMICS

**Runtime comparison between neural-ODEs and a traditional ODE solver.** Here the goal is to evaluate the neural-ODE' inference time to produce the generators' state variables estimates and to compare it with the computational time of a traditional ODE solver. Given the synchronous generator model described by (12), a numerical ODE solver could be adopted to determine the evolution in time of the state variables $\delta^g(t)$ and $\omega^g(t)$. However, in case of unstable conditions, the system response can be as rapid as, or even exceed, the time required for computing the ODE solution with a numerical solver. This situation is depicted in Figure 6 where unstable conditions arise before a numerical solution to the system of differential equations (12) is computed. Conversely, the neural ODE model $\mathcal{N}_\phi^g$ is capable of detecting unstable conditions before the system transitions into an unstable state, while also providing a good approximation of the solution. This speed advantage arises

Table 3: Average (steady-state) decision errors for the WSCC 9-bus system across different approaches based on 40 trials.

| Models | | MSE (Mean Squared Error) $\times 10^{-4}$ | | | |
|---|---|---|---|---|---|
| $\mathcal{F}_\omega$ | $\mathcal{N}_\theta$ | $p^r$ | $q^r$ | $|V|$ | $\theta$ |
| DE-OP (ours) | | $2.45 \pm 0.253$ | $3.26 \pm 0.127$ | $2.55 \pm 0.354$ | $3.82 \pm 0.924$ |
| MSE | $\emptyset$ | $1.90 \pm 0.272$ | $1.63 \pm 0.580$ | $0.32 \pm 0.153$ | $0.43 \pm 0.149$ |
| LD | $\emptyset$ | $1.77 \pm 0.163$ | $1.72 \pm 0.284$ | $0.16 \pm 0.099$ | $0.55 \pm 0.051$ |
| DC3 | $\emptyset$ | $1.86 \pm 0.217$ | $1.65 \pm 0.262$ | $0.26 \pm 0.195$ | $0.48 \pm 0.343$ |

Table 4: Average (steady-state) decision errors for the IEEE 57-bus system across different approaches based on 40 trials.

| Models | | MSE (Mean Squared Error) $\times 10^{-3}$ | | | |
|---|---|---|---|---|---|
| $\mathcal{F}_\omega$ | $\mathcal{N}_\theta$ | $p^r$ | $q^r$ | $|V|$ | $\theta$ |
| DE-OP (ours) | | $5.05 \pm 0.175$ | $7.42 \pm 1.482$ | $2.99 \pm 0.214$ | $4.43 \pm 0.673$ |
| MSE | $\emptyset$ | $3.48 \pm 0.321$ | $3.86 \pm 1.512$ | $0.77 \pm 0.148$ | $1.42 \pm 0.237$ |
| LD | $\emptyset$ | $3.97 \pm 0.279$ | $3.52 \pm 2.427$ | $0.34 \pm 0.012$ | $0.95 \pm 0.054$ |
| DC3 | $\emptyset$ | $3.31 \pm 0.579$ | $6.74 \pm 0.580$ | $0.51 \pm 0.078$ | $0.64 \pm 0.081$ |

Table 5: Average and standard deviation of constraint violations and (steady-state) optimality gap on the WSCC 9-bus system for different approaches based on 40 independent runs.

| Models | | Metrics | | | |
|---|---|---|---|---|---|
| $\mathcal{F}_\omega$ | $\mathcal{N}_\theta$ | Stability Vio. | Flow Vio. $\times 10^{-3}$ | Boundary Vio. $\times 10^{-4}$ | Optimality Gap* (at steady-state) |
| DE-OP (ours) | | $0.00$ | $8.32 \pm 0.596$ | $0.41 \pm 0.243$ | $0.13 \pm 0.02$ |
| MSE | $\emptyset$ | $2.26 \pm 0.189$ | $10.45 \pm 2.183$ | $9.72 \pm 4.930$ | $0.13 \pm 0.02$ |
| LD | $\emptyset$ | $2.13 \pm 0.175$ | $7.19 \pm 0.425$ | $0.00$ | $0.11 \pm 0.01$ |
| DC3 | $\emptyset$ | $2.45 \pm 0.205$ | $0.00$ | $0.00$ | $0.11 \pm 0.01$ |

from the neural-ODE' vector field approximation of (12), which enables quicker computation of the forward pass of a numerical ODE solver (Kidger, 2022). Table 2 reports the average and standard deviation of computational time, for numerical solvers, and inference time, for neural-ODEs, given 2 different numerical algorithms. *Neural-ODE models are, on average, about* 20 *times faster than a numerical solver which uses the dynamic equations of (12). This aspect makes neural-ODE models natural candidates as dynamic predictors for the generator model in real-time applications.*

**Comparison between neural-ODEs and PINNs.** Here the goal is to asses the precision of the neural-ODEs' estimate of the generator state variables and to compare them with PINNs Misyris et al. (2019). PINNs are ML-based models that incorporates known physical laws into the learning process. Instead of relying solely on data, PINNs use physics-based constraints to guide the training, ensuring that the model's predictions are consistent with the underlying scientific principles. Figure 7 shows the neural-ODE and PINNs' state variables estimates in case of stable conditions. While a neural-ODEs model produces highly accurate state variables' predictions, a PINN model trained on the same dataset $\mathcal{D}^g$ but affected by a generalization bias, is incapable of capturing the generator dynamics across different instances of the generator model (12) and produces poor state variables estimates. Specifically, the percentage $\ell^2$ error between the numerical ODE solver solutions $\delta(t), \omega(t)$ and the neural-ODE (NODE) predictions $\delta_{\text{NODE}}(t), \omega_{\text{NODE}}(t)$ is 5.17%, while for the PINN predictions $\delta_{\text{PINN}}(t), \omega_{\text{PINN}}(t)$ is significantly higher at 69.45%.

### D.2 Constraint violations and (steady-state) decision errors

Tables 3 and 4 report the (steady-state) decision error (MSE) at test time of DE-OP and the baseline PO methods on the WSCC 9-bus and IEEE 57-bus system, respectively. Specifically, the tables reports the MSE of estimated solutions with respect to the ground-truth solutions, with the assumption that the synchronous generators are in steady-state. The same considerations reported in Section 5.2 regarding the steady-state optimality gaps apply also for the following discussion. In other words, the ground truth variables used to compute the decision errosr, are obtained from solving the *steady-state* ACOPF problem, and as such, the decision errors here reported do not necessarily reflect that of each method for solving the Stability-Constrained AC OPF problem. Nonetheless, this metric provides valuable insights on the impact of the decision variables on the dynamics requirements of Problem 2.

Firstly, note that, for each test case, all the methods achieve similar decision error. Despite that, as shown in Table 1 of the main paper and Table 5, and Figures 4, 5 of the main paper, DE-OP is the only method that satisfy exactly the dynamic requirements (17), while all the baseline methods systematically violate the stability constraint. These results suggest that DE-OP modifies potentially unstable set points, at a cost of a only slightly higher MSE than the baseline approaches. Note in particular the MSE error of the OPF variables $|V|$ and $\theta$; these variables directly affect the generator dynamics in (12), and thus their modification is necessary to satisfy the stability constraint. This trade-off is crucial for practical applications, where the dynamic requirements must be addressed. Table 5 shows the violation of the static (flow and boundaries), along with the optimality gap with the assumption that the generators are in steady-state, for each method on the WSCC-9 bus system. Similarly to the IEEE 57 test-case discussed in Section 5.2, DC3 is the only method which achieves steady-state constraint satisfactions. All methods except DC3 generate comparable violations of the flow balance constraints, which is the most difficult constraint to satisfy due to its non-linear nature defined by Constraint (11f) and (11g). LD satisfy the boundary constraint by projecting its output $\hat{u}$ within the feasible set defined by Constraints (11b)-(11c). Empirically, we found that removing this projection operation within the DE-OP model $\mathcal{F}_\omega$, in some cases allows to satisfy the dynamic requirements. We did not thoroughly investigate this result, but our intuition is that in some cases the decision variables $|V|$, $\theta$ involved in the stability analysis must assume values close to their boundaries to satisfy stability constraints. This comes at a cost of minimal boundary constraint violations from DE-OP. MSE, lacking of a mechanism to encourage constraint satisfactions, produces solutions violating each constraint function.

### D.3 Steady-state optimality gaps

This subsection discusses the sub-optimality of the estimated solution $\hat{u}$ with respect to the ground truth $u^\star$, with the assumption that the generators are in steady-state conditions, given parameters $\zeta$ and measured in terms of objective value (16a), of DE-OP and each baseline method. The same considerations reported in Section 5.2 regarding the steady-state optimality gaps and how this metric should be interpreted, apply also for the subsequent discussion. Table 1 in the main paper and Table 5 report the optimality gaps on the WSCC-9 and IEEE-57 bus system, respectively. The tables report that all the methods achieve comparable gaps on each test case. This is intuitive, since the optimality gap depends solely on the power generated $p^r$ (see objective function (16a)), and all methods produce similar $p^r$'s prediction error, as displayed in Tables 3 and 4. For the WSCC bus system, DE-OP produces average optimality gap of $0.13\%$ while preserving system stability, that are comparable with the best optimality gap - LD with $0.11$ and DC3 with $0.11$ - which often violates stability constraints.

### D.4 Inference time

Finally, we evaluate the average inference time of DE-OP and each baseline proxy optimizer method. Table 6 shows the inference time of each proxy optimizer method on each test case. On average, DE-OP produces near-optimal and stable solutions in $1$ (ms) and $9$ (ms) for the WSCC-9 bus and IEEE 57-bus, respectively, which is slightly higher but comparable with the MSE and LD approaches, and about $15\times$ faster than the DC3 method. DC3 achieves the highest inference time, due to its correction and completion procedure, which requires solving a nonlinear system of equations and the Jacobian matrix computation. While DE-OP can be already used for near real-time applications,

Table 6: Average and standard deviation of inference times for different OPF learning approaches in the test cases.

| Models | | WSCC 9-bus | IEEE 57-bus |
|---|---|---|---|
| $\mathcal{F}_\omega$ | $\mathcal{N}_\theta$ | Inference Time (sec) | |
| DE-OP (ours) | | $0.001 \pm 0.00$ | $0.009 \pm 0.00$ |
| MSE | $\emptyset$ | $0.000 \pm 0.00$ | $0.001 \pm 0.00$ |
| LD | $\emptyset$ | $0.000 \pm 0.00$ | $0.001 \pm 0.00$ |
| DC3 | $\emptyset$ | $0.025 \pm 0.00$ | $0.089 \pm 0.00$ |

its efficiency could be improved by computing the state variables in parallel, since each dynamic predictor is independent. *This aspect makes DE-OP' inference time independent from the size of number of state variables and dynamical systems, suggesting potential for large-scale and complex systems.*

# E    ADDITIONAL EXPERIMENTAL RESULTS: DYNAMIC PORTFOLIO OPTIMIZATION

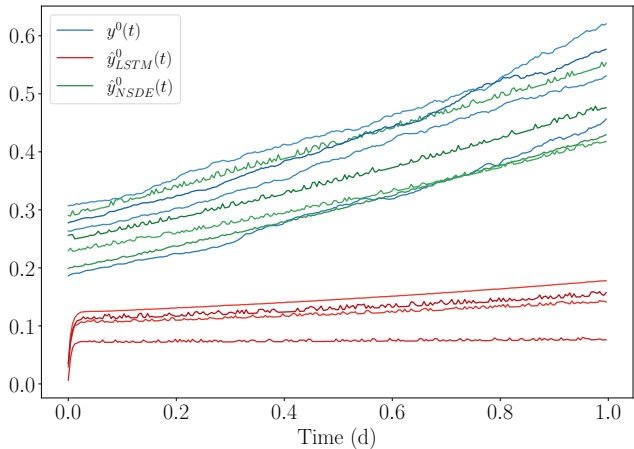

Figure 8: Asset prices (blue), LSTM (red) and neural-SDE asset prices estimates.

## E.1    LEARNING THE ASSET PRICE DYNAMICS

**Comparison between neural-SDE and LSTM.**    Fig. 8 illustrates the asset price trends given different initial asset prices $\zeta^0$, with estimates $\hat{\boldsymbol{y}}^0(t)$ provided by both a neural-SDE model and an LSTM model, alongside the true asset prices $\boldsymbol{y}^0(t)$ computed with a numerical SDE solver implementing Euler-Maruyama method (Hu et al., 2018). It is evident that, given different initial asset price $\zeta^0$, the neural-SDE model produces more accurate predictions than the LSTM model, by explicitly capturing the asset pricing dynamic equations. These accurate predictions lead to more informed and higher quality decision making, as discussed in Section 5.1.

## E.2    OPTIMALITY GAPS

This section report additional results of DE-OP and the baseline methods across different proxy optimizer methods and asset price predictors on the Dynamic Portfolio Optimization task with $n = 20$ and $n = 50$. Figures 9 and 10 display the average, percentage *optimality gap* on the test set, across different methods, for $n = 20$ and $n = 50$, respectively. In both figures, it is evident that for a given proxy optimizer method (e.g., Lagrangian Dual), by using neural-SDE predictors to capture the asset prices dynamics, DE-OP yields superior decision quality compared to the baseline methods. For $n = 20$, DE-OP achieves the lowest optimality gaps - 12.92% for DC3, 5.23% for

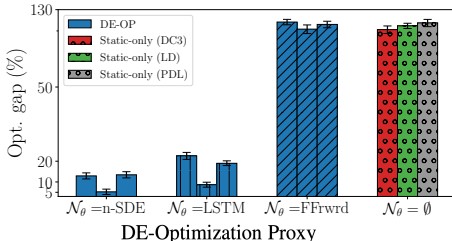
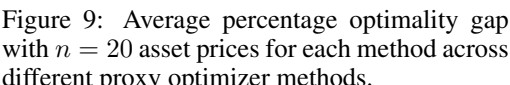

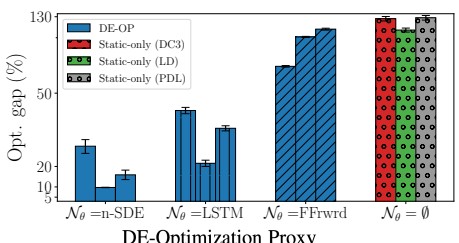

Figure 9: Average percentage optimality gap with $n = 20$ asset prices for each method across different proxy optimizer methods.

Figure 10: Average percentage optimality gap with $n = 50$ asset prices for each method across different proxy optimizer methods.

LD, and $13.45\%$ for PDL - by capturing the asset prices dynamics via explicit modeling of the asset prices' dynamics. Predicting the final asset price with LSTM leads to optimality gaps of $8.67\%$ for LD, and $19.00\%$ for PDL for PDL, performing consistently worse than DE-OP, due to its lack of explicit dynamic modeling. The Feed Forward model performs significantly worse, leading to significantly higher gaps - $121.56\%$ for DC3, $100.45\%$ for LD, and $110.33\%$ for PDL - highlighting its limitations in capturing the time-dependent nature of the data, similarly to the proxy optimizer methods which ignore the system dynamics, which achieve $103.98\%$ for DC3, $111.63\%$ for LD, and $115.11\%$ for PDL. Overall, these results follow the trend reported in Figure 3 and discussed in Section 5.1 of the main paper, concerning the optimization task with $n = 50$. Among the proxy optimizer methods considered, Lagrangian Dual consistently outperforms DC3 and PDL, suggesting that precomputed solutions can enhance the accuracy and robustness of optimization proxies. The optimality gaps achieved by each method when $n = 50$, increase with respect to the optimality gaps achieved by the corresponding method when $n = 20$, likely due to a higher complexity of the optimization task. These results highlight the importance of accurate dynamic predictions, which in turn enable, in a subsequent stage, generating high quality investment allocations.

