# OpenReview forum: "Learning to Solve Differential Equation Constrained Optimization Problems"
_ICLR.cc/2025/Conference — ICLR 2025 Spotlight_

### Official Review · Reviewer_Gqd7 · 2024-10-29

**Soundness:** 2
**Presentation:** 3
**Contribution:** 2
**Rating:** 6
**Confidence:** 3

**Summary:**

This paper introduces the Differential Equation Optimization Proxy (DE-OP) for solving differential equation (DE)-constrained optimization problems. The major contribution is a proposed primal-dual framework to ensure constraint satisfaction while allowing end-to-end differentiation. Experiments on several real-world background optimization problems show that the DE-OP achieves improvements in solution stability compared with existing static methods.

**Strengths:**

* The paper is well-structured and presented in a clear academic style.
* The experiment results demonstrate that the DE-OP can be 25 times faster than existing methods, which is a great improvement.
* Benchmarks are chosen as some real-world scenarios (AC-Optimal power flow optimization and dynamic portfolio optimization), which are beneficial for actual applications.
* The constrained optimization with differential equations is quite important for real-world control problems.

**Weaknesses:**

* Contribution 2 appears to be more of a methodological explanation of DE-OP rather than a clear, original contribution. Clarifying its novelty could help highlight its value within the paper.
* The description of the challenges addressed by DE-OP could be more specific. Currently, DE-OP mentions a wide range of problems such as NP-hard, high dimensionality, and nonlinearity that are common to many “learning-to-optimize” frameworks. A detailed description of the unique challenges faced by DE-OP would make the problem more specific.
* The proposed DE-OP is similar to many existing optimal control networks, such as the actor-critic model. Provide more details and comparisons between them or at least discuss the differences between DE-OP with neural networks for solving differential equations.
* For equation (6), when L^{DE-OP} is minimized, how does DE-OP ensure constraint satisfaction? This is not explained in the main text, and Appendix C.1 only deals with the proxy optimizer approach, leaving this key detail ambiguous.
* In the experimental setting, baselines focus on “steady-state” aspects, raising questions about whether these baselines can effectively address the problem as formulated in Equation (1). This focus also raises potential fairness concerns in the comparison. Authors could either justify their choice of baselines more thoroughly, or include additional baselines that are more directly comparable to DE-OP in terms of handling dynamic aspects of the problem.
* The results in Table 1 do not strongly support DE-OP's superiority over baselines, as only stability violations are minimized, while other metrics do not place DE-OP even as the second-best method.  Please provide more context or discussion around these results. For example, you could explain why minimizing stability violations is particularly important in their problem setting or discuss the trade-offs between different performance metrics.

**Questions:**

* Is the upper bound on the number of points a typo? It seems should be T rather than \top
* It is hard to follow the paragraph "Optimizing over the distribution of instances". Why are the parameters \zeta used as input of the neural network?
* How is the dataset D acquired for the experiment?
* Why is DE-OP compared to LSTM models in the experiments? Recurrent neural networks are generally unsuitable for DE-constrained optimization, so the rationale behind this comparison is unclear. Provide a clear justification for including LSTM models in the comparison. If there's a specific reason for this choice, please explain it.

---

> ### Author Response · Authors · 2024-11-17
> **Authors response (pt. 1)**
>
> Thank you for the thorough review and insightful feedback. We have carefully considered each of your comments and questions and tried to address them comprehensively.
>
>
> **Contribution and novelty**
>
> Thanks for the opportunity to clarify.
>
> On contribution 2: the paper shows the ignoring the system dynamics leads to decision which often violate dynamic requirements. This has not been shown in previous proxy optimizers works, which main focus is on steady-state optimization tasks.  Please also note that the DE-OP framework we propose introduces a unique new approach by combining a primal-dual formulation with end-to-end differentiability, ensuring constraint satisfaction during optimization for this challenging class of problems. This capability allows DE-OP to bridge the gap between control strategies and system dynamics, enabling efficient, dynamic optimization in contexts including energy optimization and market decisions, like we showcase in our paper.
>
>
> **Detailed Description of Challenges Addressed by DE-OP**
>
> Our proposed framework is designed to tackle a generic DE-constrained optimization problem, which can present non-linearities and non-convexity both in the system dynamics and optimization component. In the experiments we tackle a 1) DE-constrained optimization problem with 128 decision variables, 4 system of non linear ODE, each with 2 state variables and non convex constraints; we refer to Appendix B, Model 2 for the problem formulation. A 2) DE-constrained optimization problem with 50 decision variables, 50 state variables, quadratic objective and stochastic system dynamics. The diversity of these tasks highlights the wide range of applications for DE-OP. While proxy optimizer methods have successfully tackled several NP-hard problems, they do not consider dynamic aspects. In addition to have DE constraints, our methodology can also be applied to settings in which the system dynamics are coupled with the optimization component (Appendix B, Model 2) , which add additional complexity to the problem setting, as illustrated in Figure 2 and discussed in Section 4.
>
> We have tried to make these challenges explicit in the subsection titled "Challenges" under Section 3, but,
> if the reviewer suggests, we can revise these statements and further elaborate on these unique challenges.
>
> **Comparisons with Existing Optimal Control Networks**
>
> You mention that the proposed DE-OP framework may appear similar to existing actor-critic models, and suggest that more details or comparisons could clarify this relationship. While both DE-OP and actor-critic frameworks aim to optimize control strategies, their underlying motivations are fundamentally different. Actor-critic models are typically rooted in reinforcement learning, where the actor learns policies based on feedback from the critic, and not common to the setting studied in this work, which deals with solving  constrained optimization problems. Unlike actor-critic models, DE-OP solves optimization problems constrained by differential equations. Its primary focus is on maintaining feasibility throughout the optimization process via a differentiable proxy and it achieves this by embedding system dynamics directly into its dual-network structure. Also, unlike actor-critic models, which adaptively learn policies over discrete steps, DE-OP continuously solves dynamic, constrained optimization problems where system states evolve based on control variables.
>
> We don't necessarily see a connection between these areas, but if this is strongly suggested, we can add a paragraph in the Related Work section comparing DE-OP with actor-critic models, highlighting differences in architecture, training methods, and application focus.

---

> ### Author Response · Authors · 2024-11-17
> **Author response (pt. 2)**
>
> **Ensuring Constraint Satisfaction in Equation (6)**
>
> You ask how DE-OP ensures constraint satisfaction when minimizing $L^{\text{DE-OP}}$. The primal-dual structure of DE-OP effectively emulates a dual ascent method in which Lagrange multipliers are updated to enforce  constraint satisfaction during the optimization process. To elaborate, this is a max min problem in which the Lagrange multipliers values associated with constraint violations are maximized, while the Lagrangian dual objective is minimized. The primal dual process is effectively conducted with a gradient descent method (so this is seamlessly integrated in the training procedure). Notably, this allow us to handle dynamic constraints into DE-OP model training, as illustrated in Figure 4 and 5, a key aspect in our framework.
> This is an important contribution of the paper, thus, while Appendix C.1 discusses the proxy optimizer approach in details, we'll add an explicit subsection in Section 3.3 that further details how the primal dual network handles constraints throughout training. This addition will link the dual step update (the last equation before section 5) to the DE-OP constraints defined in Equation (1) during optimization. We hope this clarifies your question.
>
> **Justification of Baselines Used in Experiments**
>
> You raise a point about the use of "steady-state" baselines. Recall that our work focuses on solving DE-constrained optimization problems in near-real time and note that there may not be a practical solution for real-time joint handling of non-linear optimization and system dynamics.  This real-time requirement makes it natural to compare our approach with proxy optimizer methods (current state of the art techniques). Similarly to DE-OP, they offer very fast inference times, as shown in Appendix D, Table 6. Exact approaches, such as direct collocation and shooting methods, are instead highly impractical for real-time applications, as they require discretization of the independent variable and solving a system of algebraic equations at each interval. Additionally, the computational complexity associated with those methods increases when the system dynamics are complex and nonlinear, as in the case considered and addressed in our experiments.
>
> Please also note that in power system, for example, current industry standard handle instabilities (due to system dynamics) greedly using mechanical automatic droop control or automatic generation control rather than being included in the overall optimization process. We'll add a note to further emphasize the limitations of the current techniques when applied to dynamic problems in the revised paper.
>
> **Results in Table 1 and Discussion on Stability Violations**
>
> You comment that the results in Table 1 do not strongly support DE-OP's superiority over baselines. This is not completely correct however. The analysis here focused on minimizing stability violations; this is critical in DE-constrained optimization where maintaining constraint feasibility is essential for practical applications (e.g., power grid operations and financial risk management). Stability violations can lead to significant downstream issues that outweigh performance trade-offs in other metrics. Note that the "optimality" metrics also have an '*' sign: they are not to seen as an apple-to-apple comparison because the baselines do not enforce stability constraints and can therefore produce objectives whose costs are lower than the objectives found by our approach. However, as shown in our analysis, these baseline objectives correspond to unfeasible solutions!  In the revised version, we will expand further the discussion in Section 5.4 to explain why our analysis is important and significant in the practical contexts analyized.
>
> **Q: Clarification on Upper Bound Notation**
>
> You ask if the notation for the upper bound on the number of points, represented as $\top$, is a typo and suggest it should be $T$. This is indeed a notation issue, and we appreciate your attention to detail. We will correct this notation in the final manuscript to avoid confusion.

---

> ### Author Response · Authors · 2024-11-17
> **Authors response (pt. 3)**
>
> **Q: It is hard to follow the paragraph "Optimizing over the distribution of instances"**
>
> Parameters $\zeta$ model scenarios that influence the optimization process, such as different market states in portfolio optimization or varying demand levels in energy systems. Recall that DE-OP learns approximate optimal solutions $u^*$ from input data (problem parameters $\zeta$). We focus on solving distinct problem instances induced by a distribution of problem parameters $\zeta$.
> This approach enables the network to generalize across a distribution of possible problem instances, and is typical when learning surrogate optimization models.
> To clarify the role of $\zeta$ we will revise the paragraph to provide a clearer explanation and additional context.
> We can also include an illustrative example, if the reviewer deems it useful, in Section 4.1.
>
>
> **Q: Dataset Acquisition**
>
> The datasets are constructed using real-world data sources specific to each application. They include historical power flow data for AC-Optimal power flow optimization and financial market data for portfolio optimization.
> In more details, for the dynamic portfolio experiment, data is obtained from the Nasdaq database to form initial asset prices, while asset price dynamics and final asset prices are generated using a SDE solver (please see paragraph 'Datasets and methods', Section 5.1). For the energy experiment, the nominal loads are obtained from the benchmark library Power Grid Lib (which can be found at https://github.com/power-grid-lib/pglib-opf) and are reflective of real loads. These loads are then are perturbed to generate a distribution of  demands, using methods proposed in previous studies (please refer to paragraph 'Datasets', Section 5.2).
>
> **Q: Justification for Comparing DE-OP to LSTM Models**
>
> You question the rationale for comparing DE-OP to LSTM models. In time-series LSTMs are commonly used and provide extremely good estimators. Here, however, the comparison with LSTM is useful to highlight the role of different asset price predictors on the _final decision quality_.
> Figure 8 in the Appendix illustrates that LSTMs produce less accurate predictions of asset prices compared to neural-SDE. This is because LSTMs treat the data solely as a time series, overlooking the underlying dynamics. In contrast, neural-SDE explicitly models the asset pricing dynamic equations, resulting in more accurate predictions.
> This predictive discrepancy translates in DE-OP with neural-SDE producing superior decision quality than DE-OP with LSTM, as shown in Figure 3 of the main paper.
>
> ---
> Thank you for your feedback and insightful questions. We will incorporate these clarifications and revisions to highlight DE-OP's novel contributions and ensure the results are contextualized. We believe these improvements will address any misunderstandings and further showcase the paper's impact and rigor, and hope they could potentially lead to a reevaluation of your score.

---

> ### Comment · Reviewer_Gqd7 · 2024-11-18
> **Feedback on authors repsonse**
>
> I think most of my concerns have been well addressed, well done! I will raise my score to 6, considering the importance of differential equations and constrained optimization. Hopefully, the details you promise to add will improve the manuscript, preferably before publication. Still, more efforts can be put into finding more general and efficient solutions for optimization with differential constraints.

---

> > ### Author Response · Authors · 2024-11-18
> >
> > Thank you for your positive response. We’re pleased that our revisions addressed your concerns and highlighted the significance of our contributions. We also agree about the significance of this area and are excited about this project's outputs. Of course, we'll ensure that the revisions will be done before publication (as a matter of fact, we have already started incorporating them).
> >
> > If any further clarifications could be added, please let us know. We'd like to have your full support. Again, thank you for the thoughtful review.

---

### Official Review · Reviewer_ZAzm · 2024-11-02

**Soundness:** 2
**Presentation:** 3
**Contribution:** 3
**Rating:** 8
**Confidence:** 3

**Summary:**

The paper proposes a novel method for solving optimization problems constrained by differential equations (DEs). This method combines neural networks with a proxy optimization model, employing a dual-network architecture: one network approximates control strategies, while the other solves the differential equations related to these decisions. Experiments have validated the effectiveness of this method in fields such as financial modeling and energy optimization. Compared to traditional proxy optimization methods, this approach shows higher accuracy when handling high-dimensional dynamic constraints.

**Strengths:**

1.	The writing is clear and well-structured, making complex concepts accessible.
2.	The methodology is rigorous and well-executed, with robust experiments that provide reliable results.
3.	The authors provide thorough comparisons with existing optimization methods, demonstrating significant performance improvements.

**Weaknesses:**

1.	The authors do not adequately address potential limitations or challenges faced when applying their method in practice.
2.	The absence of diagrams or flowcharts makes complex concepts challenging to follow. Adding a diagram illustrating the dual-network architecture, particularly the interaction between the proxy optimization model F_\omega  and the solver N_\theta significantly improves understanding.

**Questions:**

1. Does the dual-network architecture need to consider synchronization issues during training? When one network changes, can the other network quickly adjust to adapt to the new state?
2. The authors have not sufficiently addressed the practical challenges of applying their method. Key aspects such as computational resource requirements, scalability to larger systems, and performance in the presence of noisy or incomplete data require comprehensive discussion.
3. The paper does not specify the numerical solvers used for comparison. In equations (3a) and (3b), where N_\theta approximates the evolution of state variables, how does the performance of the neural network solver compare to traditional numerical solvers like Runge-Kutta or Euler methods? A comparison focusing on accuracy, efficiency, and convergence is needed for completeness.
4. The loss function integrates the optimization of state and control variables, but how are the contributions of these two networks balanced?
5. The proposed network should discuss its numerical stability during the solving process, especially in cases where F is a highly nonlinear or stiff equation. How does the network avoid issues of gradient explosion or vanishing?
6. The discussion on Neural ODE in the abstract should include a reference to the paper by Chen et al. (2018) titled "Neural Ordinary Differential Equations," presented at NeurIPS, to provide necessary background and support for the proposed method.

---

> ### Author Response · Authors · 2024-11-17
> **Authors response (pt. 1)**
>
> **Q1: Synchronization Issues During Training**
>
> You ask whether the dual-network architecture considers synchronization issues during training and how quickly one network can adapt when the other changes. This is a crucial aspect of our approach and during our experiments, we did not observe any synchronization issues during training. We employed alternating gradient updates, where each network trains iteratively to ensure that the adjustments made by one network are complemented by the other. It is important to remark that the neural-DE models can be initialized to produce accurate estimates of the state variables, which is a key to leverage good approximation of the optimal decision variables ${u}^\star$ as the optimization progresses.
>
> In the revised version, we will expand the training procedure details in Section 4.1 to include an explanation of how synchronization is maintained during training. Additionally, we will discuss techniques such as gradient accumulation and adaptive learning rates that could employed to enhance the responsiveness and stability of both networks during updates.
>
>
> **Q2: Numerical Solvers Used for Comparison**
>
> You inquire about the performance comparison between the neural network solver and traditional numerical solvers. We indeed benchmarked against solvers emplying established numerical algorithms such as Dopri5 and Bosh3, which are widely used for solving differential equations in high-dimensional spaces. The performance of our neural network solver, denoted as $\mathcal{N}_\theta$, was evaluated based on computational efficiency and accuracy against these solvers for ordinary differential equations and reported in the paper. Please refer to Appendix D: Figure 6, Figure 7 and Table 2.
>
> We also reported a comparison between neural-SDEs and a traditional numerical solver for stochastic differential equations. Please refer to Appendix E and Figure 8. We hope this clarifies your question. We will also report these additional details in the main corpous of the paper, in our revised version.
>
>
> **Q3: Balancing Contributions of the Dual Networks in the Loss Function**
>
> The point on the simultaneous optimization of decision and state variables is important. The contributions of our dual network architecture is balanced using a Lagrangian Dual framework, which incorporates constraint violations into the training process. This approach allows us to handle both static and dynamic constraints by associating them with Lagrange multipliers $\lambda$. Note that the $\lambda$ terms are not hyperparameters, but are updated directly in the dual training approach, using a "neural" equivalent of a dual ascent method, thus encouraging satisfaction of the constraint functions while maintaining alignment with the overall objective function.
>
> In the revised version, we will further clarify this important aspect, which we believe is a key strenght of our approach. We will also show how these dual weights are optimized during training.
>
> **Q4: Numerical Stability with Highly Nonlinear or Stiff Equations**
>
> You raise a valid concern regarding numerical stability, especially in cases where $F$ represents a highly nonlinear or stiff equation. We have taken several measures to address potential issues like gradient explosion or vanishing (e.g., using standard gradient clipping and adopative step-size solvers) but our experiments **did not** report any particular challenge in this regard.
>
> Note also that our primary focus is on the inference stage, where we also did not observe any particular challenges in this regard.
>
> **Q5: Reference to Neural Ordinary Differential Equations (Chen et al., 2018)**
>
> Thank you! We will revise the abstract and the Related Work section to include this reference.
>
>
> **W1. Potential Limitations and Challenges in Practical Applications**
>
> You comment on the absence of a discussion about potential limitations or challenges in applying our method in practice. However, it is not entirely clear what specific practical limitations you are referring to. Could you elaborate further on this point? Our experiments already include a diverse set of real-world scenarios, using actual data from domains such as financial modeling and energy optimization. These cases were selected to reflect practical application challenges, including handling high-dimensional constraints and non-linearities in dynamic systems. The robustness of our method has been demonstrated through comprehensive comparisons with existing approaches, and we believe this has highlighted its effectiveness in both computational performance and accuracy.
>
> To better address your feedback, we would appreciate further detail on which specific challenges you believe warrant additional discussion. In response, we can extend our Discussion section to include these practical considerations and our strategies for overcoming them in real-world applications.

---

> ### Author Response · Authors · 2024-11-17
> **Authors response (pt. 2)**
>
> **W2: Inclusion of Diagrams or Flowcharts**
>
> We appreciate this suggestion of including a flowchart and we also would like you to refer to our Figure 2, which was created with this goal in mind. Does the reviewer have additional suggestions for us to improve it? We would surely greatly appreciate them!
>
>
> --
> Thank you for your constructive feedback. We have outlined revisions that will enhance the clarity, depth, and practical insights of our paper and hope these changes will make our contributions and methodologies clearer. We hope these improvements will demonstrate the significance and quality of our work, potentially supporting a stronger consideration of the evaluation score.

---

> > ### Author Response · Authors · 2024-11-21
> >
> > Hello, reviewer ZAzm. As the discussion period is underway,  we wanted to check if our response has addressed all your concerns.  Please let us know if further explanation would be helpful.
> > Many thanks!

---

> > > ### Author Response · Authors · 2024-11-23
> > >
> > > Hello, reviewer ZAzm. As the discussion period is near ending and we haven't heard from you yet, we wanted to check again if you have pending questions. Please let us know, we are happy to provide additional explanations as needed.
> > > Many thanks!

---

> > > > ### Comment · Reviewer_ZAzm · 2024-12-03
> > > >
> > > > Thank you for your response and detailed explanation. I am pleased to see that you have well addressed the concerns I previously raised. Your clarifications and improvements have significantly enhanced the clarity of the work. As a result, I have updated my score to 8.

---

> > > > > ### Author Response · Authors · 2024-12-03
> > > > >
> > > > > We are happy that our explanation and revision addressed your concerns. Thank you again for your time and review. This is greatly appreciated.

---

### Official Review · Reviewer_PcqT · 2024-11-04

**Soundness:** 4
**Presentation:** 4
**Contribution:** 3
**Rating:** 8
**Confidence:** 4

**Summary:**

In this paper the authors propose a dual learning approach for solving differential equation-based optimization problems, relevant to dynamic optimization and systems control. The authors present a financial and a power-grid case study, showcasing the accuracy and speed of their proposed method. The method includes several novel aspects and its applicability is well-motivated by the authors.

**Strengths:**

- Well-written paper, well organized and well-motivated.
- The selected case studies are important, challenging and sufficient to show the effectiveness of the proposed approach.
- Authors have done a lot of interesting and relevant work, presenting many more comparisons and discussion in the Appendices.

**Weaknesses:**

Related works section:
- This work is relevant to control applications, which is a really rich area with respect to literature. The Reviewer realizes that it would be impossible to review all of it, but some statements and key relevant areas could be better represented. Several questions on this have been added to the Questions section.
- The motivation includes stochastic DE constrained optimization, which is not shown in results, rather briefly mentioned in Appendix A. The discussion provided in Appendix A is not sufficient to justify that this could extend to stochastic opt, which comes with more challenges.
- The authors only solve up to a 57-bus system, which is a benchmark problem in power systems optimization. Real problems are much larger than this. The authors should discuss scalability of proposed method and challenges in each step as number or nodes, variables, constraints increases.
- The authors use a local optimizer to generate data, ensuring near-optimal solutions are used to warm-start the solver. The sensitivity of the method to this is not discussed.

**Questions:**

- In the Related Works section, the authors do not mention collocation-based or differential algebraic programming (DAE), although these are mentioned later in the results/discussion section. How do these methods compare to proposed approach, why are they relevant?
- How does "multiparametric programming" (which is a field related to MPC, developed to identify "maps" of control actions under different parametrizations, for which authors have also previously used surrogate models) compare to proposed approach?
- Finally, the authors mention that DE-optimization problems involve parametrization of the state variables, and thus traditional MPC is unsuitable. This statement must be made more clear because it is a little misleading. What is traditional MPC and why is it unsuitable? Due to implementation or computational cost concerns?
- How would the method steps (training for Fw, training for Nθ), using Langrangian approach scale for larger grid systems? Please discuss this in your conclusions.
- How critical is it to the method to have near-optimal guesses to solve the optimization problem with IPOPT? Would this ever become an issue for larger systems?
- For both surrogate models trained, what is the trade-off between training sample diversity (covering the entire space) vs. optimality (samples being close to optimal regions) for accuracy and constraint violation of the overall method?

---

> ### Author Response · Authors · 2024-11-17
> **Authors response (pt. 1)**
>
> Thank you for your thorough review and the positive feedback on the motivation, methodology, and case studies of our work. We appreciate your thoughtful suggestions for improvement and questions. Below, we address each of your specific concerns:
>
> **Collocation-based methods, DAE, and multiparametric programming**
>
> Thanks for the question! Collocation-based methods solve DE-constrained optimization by discretizing the problem into a set of nonlinear equations, which can become quickly computationally expensive for systems like those studied in this paper. This is due to the dense grids needed for accuracy. DAEs are relevant as they generalize ODEs by incorporating algebraic constraints and thus suitable for problems with coupled dynamic and algebraic conditions. Importantly, our DE-OP method differs as it avoids such dense discretizations by leveraging neural networks enabling the resolution of both decision (optimization) problems and system dynamics in real-time. This is unachievable, to the best of the authors knowledge, with such traditional collocation and DAE approaches in the complex settings studied.
>
> Thanks also for the question on multiparametric programming. It provides us an opportunity to further showcase the uniqueness of our approach. While multiparametric programming is useful for scenarios with known parameter ranges, it lacks adaptability when new conditions emerge without recalculating control solutions. This is obviously unattainable in the context studied in this paper as decision making and systems dynamics are highly coupled. In contrast, our DE-OP method learns mappings directly from problem parameters using neural networks, allowing for real-time adaptability and handling dynamic, complex constraints quite effectively, as we showcase in our analysis. We will add a discussion highlighting these distinctions to enhance clarity!
>
> Finally, the statement on traditional MPC being unsuitable for DE-optimization problems will be further clarified. In fact, traditional MPC involves repeatedly solving an optimization problem over a finite horizon using current system states. This is however computationally prohibitive when the system dynamics and constraints are nonlinear and complex, and thus, it becomes unsuitable for the real-time scenarios required by our application domains.
> DE-OP circumvents these challenges by training models that approximate the control solutions and state dynamics upfront, thereby reducing the need for repeated optimization. We will emphasize this in the revised paper.
>
>
> **Stochastic DE-Constrained Optimization**
>
> We acknowledge that our discussion on stochastic DE-constrained optimization is brief, but our results indeed include such case: In Section 5.1 the paper introduces a financial modeling and optimization task (see Eq. (9)) in which the system dynamics are governed by stochastic differential equations. Here, the stochasticity in the asset price dynamics is handled by neural-SDE models, which are well suited for this type of applications.
>
> **Scalability to Larger Systems**
>
> We agree that scalability is a critical aspect for practical applications and indeed our approach was formulated with this aspect in mind! Firstly, please notice that the IEEE 57-bus represents a portion of the American Electric Power System in the U.S. Midwest region during the early 1960s. This is a common test case for optimization methods in optimal power flow problems.
> We also note that proxy optimizers methods such as [1] have shown that the proxy optimization solvers can scale up to country-wise size systems (up to 14,000 buses). Within the DE-OP setting, the dynamics of each generator are independent one another and captured by a specific neural-ODE model. Thus their computation is fully parallelized and does not represent a bottleneck in large-scale systems. We thank the reviewer for noting this point as it will allow us to further discuss this important aspect in the final version of the paper.
>
> References:
>
> [1] Mak, Terrence W. K. et al. (2023). Learning Regionally Decentralized AC Opti- mal Power Flows with ADMM. arXiv: 2205.03787 [eess.SY]. url: https: //arxiv.org/abs/2205.03787.

---

> ### Author Response · Authors · 2024-11-17
> **Authors response (pt. 2)**
>
> **Supervision and Training Flexibility**
>
> Your question regarding the necessity of near-optimal solutions during training is highly relevant. While our current approach uses a supervised framework with near-optimal solutions to warm-start the solver, the flexibility of DE-OP allows for alternative strategies. Indeed, as indicated in Section 4.3, when near-optimal solutions ${u}^{\star}$ are not available, the term $| \hat{u} - {u}^\star |$ in Eq. (6) can be replaced with $\mathcal{J}({u}, {y}(t))$ to facilitate self-supervised learning. This adaptation is current topic of research in the field developing surrogate optimization models, and has been shown very promising results [1].  We will include this insight to clarify the method's adaptability!
>
>
> **Trade-off Between Training Sample Diversity and Optimality**
>
> We appreciate the point raised regarding the trade-off between sample diversity and proximity to optimal solutions. In our finance experiment, we use the methodology described in [2] for data generation, while for the energy optimization task, we follow the approach in [3], as these are established works. These methods ensure sample diversity that covers a range of real-world scenarios. We agree that further analysis at the sample level would be beneficial in understanding the correlation between sample coverage, constraint violations, and decision quality and we will note this as an area for future work to provide more insight into how sample selection impacts method performance.
>
>
> ---
> Thank you again for your thorough review and encouraging comments. We are confident that these revisions will further strengthen our contribution.
>
>
> **References:**
>
> [1] Seonho Park, Pascal Van Hentenryck. Self-Supervised Primal-Dual Learning for Constrained Optimization. https://arxiv.org/abs/2208.09046
>
>
> [2] Sambharya, Rajiv et al. (2022). End-to-End Learning to Warm-Start for Real- Time Quadratic Optimization. arXiv: 2212.08260 [math.OC].
>
>
> [3] Fioretto, Ferdinando, Terrence W.K. Mak, and Pascal Van Hentenryck (2020). “Predicting AC Optimal Power Flows: Combining Deep Learning and La- grangian Dual Methods”. In: Proceedings of the AAAI Conference on Arti- ficial Intelligence 34.01, pp. 630–637.

---

### Author Response · Authors · 2024-11-27
**Authors update**

We thank the reviewers for their valuable suggestions and insightful feedback. We believe we have responded to all questions raised, in addition to revising the paper according to the suggestions provided. In particular, the key updates are as follows:

### As suggested by Reviewer PcqT:
- We reorganized the first paragraph of the Related Work section (lines 67-75) to provide a more structured discussion of alternative approaches for solving DE-constrained optimization problems.
- We clarified the choice of baseline methods for comparison within the setting considered in Section 5 (lines 337-339).
- We expanded Section 5.1 with a discussion on the scalability of the proposed method (lines 472-474) and, at the end of Section 4.3, outlined how the DE-OP training algorithm can be modified for self-supervised settings (lines 328-330).

### As suggested by Reviewer ZAzm:
- In Section 4.3, we added a discussion on potential synchronization issues during DE-OP model training and proposed possible mitigating strategies (lines 306-311). In the same section, we elaborated on the mutual contributions of our dual-network architecture and how these are automatically balanced during DE-OP training (lines 292-294).
- In Section 5, we clarified that the dynamic predictions are compared to numerical differential solvers, adding two sentences to emphasize this point (lines 379-380 and 459-460).
- We included a citation to the original work on neural ODEs (Chen et al., 2018) in the abstract and the Related Work section (line 90).

### As suggested by Reviewer Gqd7:
- In Section 4.3, we added a sentence to clarify how DE-OP ensures constraint satisfaction (lines 290-292).
- In Section 5.2, we expanded the discussion of the results presented in Table 1, Figure 4, and Figure 5 (lines 511-517).
- We included references in Section 5 for the sources used to generate the datasets for the experiments (line 373 and lines 462-463).

Additionally, we corrected all typographical errors throughout the paper, as also suggested by the reviewers. We hope these revisions address your comments and improve the clarity and quality of the paper. Please let us know if there are any pending questions.

---

### Meta-Review · Area_Chair_XECy · 2024-12-18

**Metareview:**

This paper proposes a dual-learning approach to solving optimization problems with differential equation constraints. The approach uses one network to map a problem-specific vector to the control solution and another network to approximate the differential equations. The training algorithm leverages steady-state solutions as a warm-up and employs a primal-dual optimization framework to enforce constraints effectively. The method’s performance is demonstrated through two case studies: one in finance and the other in power-grid optimization, highlighting its accuracy and computational efficiency.

All three reviewers provided positive feedback on this work. The paper is well-structured, written in a clear academic style, and introduces several novel aspects. The authors also provide strong motivation for the method’s applicability. Based on these strengths, the AC recommends acceptance.

**Additional Comments On Reviewer Discussion:**

The reviewers raised several concerns, including applicability to stochastic problems, computational scalability, interpretation of Table 1, dataset acquisition, and potential limitations. Most of these points were satisfactorily addressed during the discussion, and two of the three reviewers updated their scores.

---

### Decision · Program_Chairs · 2025-01-22

Accept (Spotlight)